# Differential TAM receptor–ligand–phospholipid interactions delimit differential TAM bioactivities

Erin D Lew[1], Jennifer Oh[1], Patrick G Burrola[1], Irit Lax[2], Anna Zagórska[1], Paqui G Través[1], Joseph Schlessinger[2], Greg Lemke[1,3]*

[1]Molecular Neurobiology Laboratory, The Salk Institute for Biological Studies, La Jolla, United States; [2]Department of Pharmacology, Yale University School of Medicine, New Haven, United States; [3]Immunobiology and Microbial Pathogenesis Laboratory, The Salk Institute for Biological Studies, La Jolla, United States

**Abstract** The TAM receptor tyrosine kinases Tyro3, Axl, and Mer regulate key features of cellular physiology, yet the differential activities of the TAM ligands Gas6 and Protein S are poorly understood. We have used biochemical and genetic analyses to delineate the rules for TAM receptor–ligand engagement and find that the TAMs segregate into two groups based on ligand specificity, regulation by phosphatidylserine, and function. Tyro3 and Mer are activated by both ligands but only Gas6 activates Axl. Optimal TAM signaling requires coincident TAM ligand engagement of both its receptor and the phospholipid phosphatidylserine (PtdSer): Gas6 lacking its PtdSer-binding 'Gla domain' is significantly weakened as a Tyro3/Mer agonist and is inert as an Axl agonist, even though it binds to Axl with wild-type affinity. In two settings of TAM-dependent homeostatic phagocytosis, Mer plays a predominant role while Axl is dispensable, and activation of Mer by Protein S is sufficient to drive phagocytosis.

*For correspondence: lemke@salk.edu

## Introduction

The three receptor tyrosine kinases of the TAM family–Tyro3, Axl, and Mer—are expressed in sentinel cells of the immune system, endothelial cells of the vasculature, neurons and glia of the nervous system, and professional phagocytes of the immune, nervous, and reproductive systems (*Lemke, 2013*). In these settings, TAM signaling regulates multiple functions but two are especially prominent. The first is the 'homeostatic' or silent phagocytosis of the billions of apoptotic cells (ACs) that are generated on a daily basis throughout life (*Scott et al., 2001*; *Lemke and Burstyn-Cohen, 2010*). The second is the inhibition of the innate immune inflammatory response in dendritic cells (DCs), macrophages, and other sentinel cells of the immune system (*Rothlin et al., 2007*; *Lemke and Rothlin, 2008*). Deficiencies in TAM expression or activity lead to autoimmunity, blindness, and infertility (*Lu et al., 1999*; *D'Cruz et al., 2000*; *Lu and Lemke, 2001*; *Duncan et al., 2003*; *Lemke and Lu, 2003*; *Rothlin and Lemke, 2010*; *Lemke, 2013*). Conversely, over-expression or aberrant activation of Tyro3, Axl, or Mer (gene name *Mertk*) is associated with the development, progression, and metastasis of cancers (*Avilla et al., 2011*; *Cummings et al., 2013*; *Lemke, 2013*; *Meyer et al., 2013*; *Paccez et al., 2014*). Axl in particular has been shown to mediate the resistance of solid tumors and leukemias to epidermal growth factor receptor (EGFR)-directed chemotherapeutic agents (*Hong et al., 2008*; *Zhang et al., 2012*; *Meyer et al., 2013*). Elevated TAM expression has also recently been implicated in increased susceptibility to infection by enveloped viruses (*Shimojima et al., 2006*; *Morizono et al., 2011*; *Meertens et al., 2012*; *Bhattacharyya et al., 2013*).

The importance of TAM activity notwithstanding, the differential signaling capacity of the two soluble ligands that activate TAM receptors—Gas6 and Protein S (gene and protein names Pros1)

**eLife digest** Cells send out and receive signals to communicate with other cells. Detecting these signals is largely carried out by proteins called receptors that span the cell surface membrane. These proteins typically have extracellular domains outside of the cell that can bind to specific signaling molecules and an intracellular domain inside the cell that relays the information inwards to trigger a response.

Three such receptor proteins are collectively known as the TAM receptors. Each day, many billions of cells in the human body die and are engulfed by other cells and broken down so that their building blocks can be reused. TAM receptors are required for this process; and the TAM receptors also help prevent the immune system from going out of control, which would damage the body's own tissues.

Two different signaling proteins, called Gas6 and Protein S, can bind to and activate TAM receptors. Both of the signaling proteins can also bind to a phospholipid molecule that is found on the surface membrane of dead cells. However, it is not known if all three TAM receptors bind to both signaling proteins equally, and the importance of the phospholipid-binding domain in the signaling proteins remains unclear.

To shed light on the workings of these receptors, Lew et al. created mouse cells that each only express one out of the three TAM receptors. These cells were then exposed to intact Gas6 and Protein S, or shortened versions that lacked the phospholipid-binding domain. Lew et al. found that Gas6 could trigger a response through all three TAM receptors but that Protein S was specific for only two out of the three receptors. Signaling proteins with or without their phospholipid-binding domains bound equally well to the receptors, but the maximum level of response was only triggered when both signaling proteins were intact and the phospholipid molecule was present. This is important since the phospholipid can be thought of as an 'eat-me' signal by which the dead cells are recognized by the TAM receptor-expressing cells that will engulf them.

Using mice that only produce a TAM receptor called Mer, Lew et al. show that Protein S alone can trigger the process that engulfs and breaks down cells in a living organism. These data and previous work suggest that two TAM receptors—including Mer—are involved in the daily engulfment of dying cells, whereas the third mediates this process during infection and tissue damage.

Molecules that inhibit or activate the function of TAM receptors are currently being developed to treat cancer and other diseases. By revealing which receptors respond to which signaling molecules, the findings of Lew et al. will serve to guide these efforts.

(*Stitt et al., 1995*; *Varnum et al., 1995*)—is incompletely understood. Gas6 is thought to function as a ligand for all three receptors (*Stitt et al., 1995*; *Nagata et al., 1996*; *Chen et al., 1997*), but the role of Pros1 as a ligand for one or more TAM receptors has until recently been controversial (*Godowski et al., 1995*; *Anderson et al., 2003*; *Hafizi and Dahlback, 2006*). At the same time, the unique structural features of Gas6 and Pros1—in which a C-terminal 'SHBG domain' binds to the Ig-like domains of TAM receptors while an N-terminal γ-carboxylated 'Gla domain' binds, in a $Ca^{2+}$-dependent reaction, to the phospholipid phosphatidylserine (PtdSer) (*Lemke and Rothlin, 2008*; *Lemke, 2013*)—have led to conflicting conclusions as to the relative importance of these two domains in receptor binding vs activation (*Mark et al., 1996*; *Nakano et al., 1997*; *Tanabe et al., 1997*). Although Gas6 and Pros1 binding to the PtdSer-containing membranes of enveloped viruses potentiates TAM activation (*Meertens et al., 2012*; *Bhattacharyya et al., 2013*), and the Gla domain is thought to be required to 'bridge' a TAM receptor expressed on the surface of a phagocyte to the PtdSer expressed on the surface of its AC target (*Lemke and Rothlin, 2008*; *Lemke and Burstyn-Cohen, 2010*; *Nagata et al., 2010*), a preparation of recombinant mouse Gas6 that lacks the Gla domain entirely is sold commercially as a TAM activator.

Potentially distinct roles for Pros1 and Gas6 in vivo have only recently begun to be appreciated (*Burstyn-Cohen et al., 2012*; *Carrera Silva et al., 2013*), and to date only a single genetic study of the importance of Gas6 vs Pros1 in a TAM-dependent process has been reported (*Burstyn-Cohen et al., 2012*). This study addressed TAM signaling during the phagocytosis of photoreceptor (PR) outer segments by retinal pigment epithelial (RPE) cells (*Strauss, 2005*), which express Mer and Tyro3

but no Axl (*Prasad et al., 2006*). RPE phagocytosis of the PtdSer-displaying tips of PR outer segments (*Ruggiero et al., 2012*) requires Mer—mice, rats, and humans that carry loss-of-function mutations in the *Mertk* gene are blind, due to the cell-non-autonomous degeneration of nearly all PRs (*D'Cruz et al., 2000*; *Gal et al., 2000*; *Duncan et al., 2003*; *Prasad et al., 2006*; *Mackay et al., 2010*; *Nandrot and Dufour, 2010*). This cell death results from the accumulation of toxic oxidated proteins that are generated during the course of phototransduction and are removed by phagocytosis. Retinal loss of either the mouse *Gas6* or *Pros1* gene alone yields a retina with a normal number of PRs, but the combined loss of *Pros1* and *Gas6* results in a PR degeneration phenotype that fully phenocopies the cell death seen in $Mertk^{-/-}$ mice (*Burstyn-Cohen et al., 2012*). While this analysis demonstrated that Pros1 is sufficient to drive Mer-dependent phagocytosis in RPE cells, the presence of Tyro3 in these cells raised the possibility that Pros1 activation of the Mer kinase might be dependent on Pros1 binding to Tyro3.

In the current study, we have used biochemistry, receptor activation profiling, and genetic analyses of single and compound mouse mutants to establish the basic rules for TAM ligand–receptor interaction, signaling, and function. We find that Gas6 activates all three TAM receptors, but is an especially potent ligand for Axl, with which it has a unique association. In contrast, Pros1 activates Tyro3 and Mer but is inactive as an Axl agonist. Notably, we find that the Gla domains of TAM ligands are dispensable for receptor binding but are critical for optimal receptor activation, and that for Gas6 activation of Axl, this requirement is absolute. We conclude that a complete TAM signaling module is composed of a receptor, a γ-carboxylated protein ligand, and the phospholipid PtdSer, a tripartite arrangement that is unique to the TAM family. Finally, we show that Mer is the functionally predominant TAM receptor in two different settings of 'homeostatic' PtdSer-dependent phagocytosis in vivo–in the retina and the testes–and that Pros1 binding to and activation of Mer alone is sufficient to ensure wild-type levels of phagocytosis in these settings.

## Results

### Derivation of assay tools

We first generated highly pure preparations of recombinant mouse Gas6 and Pros1. We transfected pCEP4-based expression plasmids containing cDNAs for both full-length and Gla domain-deleted ('Gla-less') versions of these TAM ligands into HEK293 EBNA cells (*Sasaki et al., 2002*) and selected stable transformants. Lines expressing the highest levels of each ligand were grown in serum-free production medium supplemented with vitamin K2, required for γ-carboxylation of Gla domain glutamic acid residues (*Bandyopadhyay, 2008*). Ligands were purified to apparent homogeneity from conditioned medium, using affinity purification on a nickel-NTA resin followed by size exclusion and anion exchange chromatography (see 'Materials and methods').

All recombinant mouse ligands ran as single bands under reducing conditions on SDS polyacrylamide gels and eluted from an anion exchange column as a single peak suggesting that the ligands were pure (*Figure 1A* and data not shown). Commercially available Protein S purified from human plasma (hPros1) ran as a single predominant band under reducing conditions, together with a secondary band of lower molecular weight (*Figure 1A*). Size exclusion chromatography suggested that full-length Gas6 in solution is a mixture of monomers, dimers, and/or higher order multimers, whereas the Gla-less form appeared as a monomer (*Figure 1—figure supplement 1*). These elution profiles are consistent with earlier work on hPros1, indicating that it forms disulfide-linked multimers, and that multimerization is enhanced by apoptotic cells (*Uehara and Shacter, 2008*).

In order to have cellular assay targets in which individual TAM receptors could be expressed and their expression level normalized between cell lines, we first prepared immortalized mouse embryo fibroblast (MEF) lines from embryonic day (E)13.5 $Tyro3^{-/-}Axl^{-/-}Mertk^{-/-}$ triple mutants ('TAM TKO') and all three possible double mutants (e.g., $Tyro3^{-/-}Mertk^{-/-}$, or 'TM DKO') (*Lu et al., 1999*; see 'Materials and methods'). The TAM TKO MEF lines provide a baseline cell population that lacks expression of any TAM receptor and the double mutant lines isolate a single receptor. This was particularly important, since in surveys of established cell lines we were unable to identify any human, mouse, rat, hamster, or monkey immortalized line that did not express at least one endogenous TAM receptor. [Selected examples of TAM expression in cultured cells are shown in *Figure 1—figure supplement 2*] This suggests that all prior analyses of TAM receptor activation have been encumbered by the expression of endogenous Tyro3 and/or Axl and/or Mer.

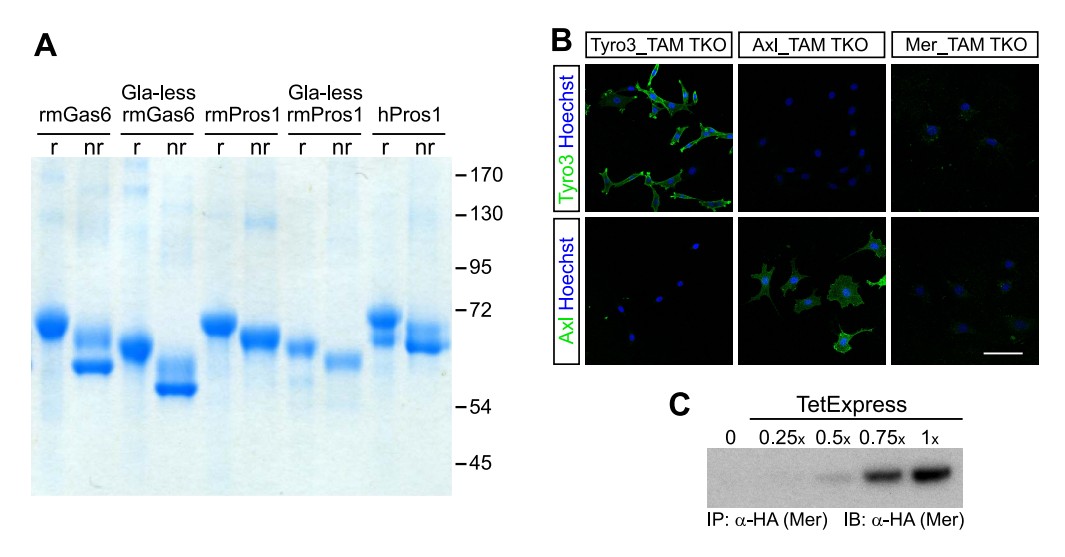

**Figure 1**. Recombinant TAM ligands and surface expression of TAM receptors. (**A**) Purified recombinant full-length and Gla-less mouse Gas6 (rmGas6 and Gla-less rmGas6, respectively) and full-length and Gla-less Pros1 (rmPros1 and Gla-less rmPros1, respectively) were run under both reducing (r) and non-reducing (nr) conditions in SDS-PAGE. In parallel, purified human Pros1 (hPros1) was also run under reducing and non-reducing conditions. Gel was stained with Gel Code Blue (Pierce). (**B**) Live cell labeling of Tyro3 (top panels) and Axl (bottom panels) on the surface of clonal populations of TAM TKO MEF lines expressing HA-tagged recombinant mouse Tyro3 (left), recombinant mouse Axl (middle), or recombinant mouse Mer (right). Bar: 100 μm. (**C**) Induction of HA-tagged Mer expression in a Mer_TAM TKO MEF line in the presence of increasing concentrations of TetExpress transactivator protein (Clontech). In this and all the subsequent blots in which the HA tag was used for both immunoprecipitation (IP) and immunoblotting (IB), two different anti-HA antibodies were used: an anti-HA high affinity for IP and an anti-HA.11 for IB.

The following figure supplements are available for figure 1:

**Figure supplement 1**. Size exclusion chromatography of full-length and Gla-less Gas6.

**Figure supplement 2**. TAM receptor expression in immortalized cell lines.

We generated multiple clonal cell lines in the background of the TAM TKO MEFs that express HA-tagged versions of mouse Tyro3, Axl, and Mer, respectively. We then selected one clonal population from the Tyro3 and Axl sets that expressed equivalent levels of these receptors, based on detection of the HA tag (*Figures 1B and 2D*; see 'Materials and methods'). Expression in these Tyro3- and Axl-expressing MEFs was not maximal among the clones we isolated but was physiological in that we did not observe receptor activation (autophosphorylation) in the absence of added ligand (*Figure 2*), a phenomenon frequently observed when RTKs are over-expressed. Although we were also able to isolate MEF lines expressing Mer, these lines always (for unknown reasons) expressed much lower levels of this receptor (*Figure 1C* and data not shown). For the Mer-expressing MEF line analyzed in this paper, we used a stable clonal line generated by expressing HA-tagged recombinant mouse Mer under the control of a TetExpress-inducible promoter (Clontech; *Figure 1C*). For the Tyro3- and Axl-expressing MEF lines, we used Tyro3 and Axl antibodies directed against the extracellular domains of the receptors to live label non-permeabilized cells in culture and demonstrate surface expression of the individual receptors (*Figure 1B*). Mer expression in clonal, Tet-inducible Mer_TAM TKO MEF lines, while detectable by western blot (*Figure 1C*), was too low to be detected by immunocytochemistry.

## Distinct TAM receptor activation profiles for Gas6 and Pros1

We first assayed full-length recombinant mouse Gas6 and Pros1, together with purified human Pros1, for their ability to activate mouse Tyro3, Axl, and Mer. We measured activation of single receptors expressed in MEFs by monitoring their tyrosine phosphorylation 10 min after ligand addition to cells

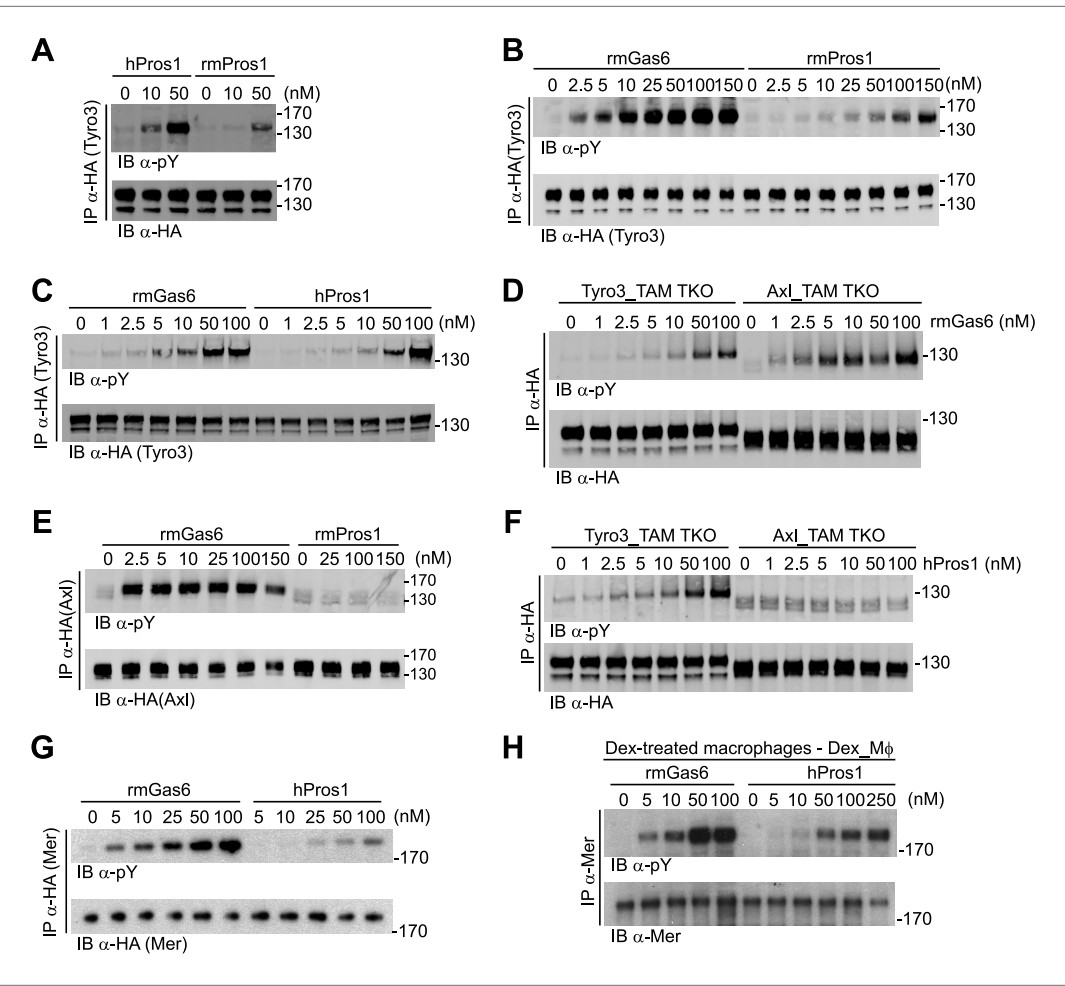

**Figure 2**. Gas6 and Pros1 exhibit TAM receptor selectivity. (**A**) Tyro3-expressing MEFs were stimulated with either human Pros1 (hPros1) or recombinant mouse Pros1 (rmPros1). (**B** and **C**) Tyro3-expressing MEFs were stimulated with increasing concentrations of full-length recombinant mouse Gas6 (rmGas6), recombinant mouse Pros1 (rmPros1, **B**), or purified human Pros1 (hPros1, **C**). (**D**) Tyro3- and Axl-expressing MEFs were stimulated with increasing concentrations of rmGas6. (**E**) Axl-expressing MEFs were stimulated with increasing concentrations of rmGas6 or rmPros1. (**F**) Tyro3- and Axl-expressing MEFs were stimulated with increasing concentrations of hPros1. (The lower molecular weight doublet in lanes 8–14 of the anti-pY blot of this panel, in lanes 1 and 8–11 of the anti-pY blot of panel **E**, and in lane 8 of the anti-pY blot of panel **D** is seen only in Axl-expressing MEFs in which Axl is not activated by exogenous ligand and its intensity does not increase with increasing concentration of hPros1 or rmPros1. It may represent basal Axl phosphorylation caused by low levels of MEF-produced (endogenous) Gas6. See also lanes 1 and 7–12 of the anti-pY blot of **Figure 3B**) (**G**) Mer-expressing MEFs were stimulated with increasing concentrations of rmGas6 or hPros1. (**H**) Dexamethasone-treated bone marrow-derived macrophages (Dex_Mφ) were stimulated with increasing concentrations of rmGas6 or hPros1. Following stimulation with ligand for 10 min at 37°C, HA-tagged Tyro3 (**A**–**D** and **F**), Axl (**D**–**F**), Mer (**G**), or endogenous Mer (**H**, figure supplement 1) were immunoprecipitated from total cell lysates and subjected to SDS-PAGE and quantitative Licor western blotting (panels **G**, **H**, ECL western blotting system) with the indicated antibodies. In this and subsequent figures, receptor activation was assessed by blotting immunoprecipitates with an anti-phosphotyrosine antibody (pY).

The following figure supplement is available for figure 2:

**Figure supplement 1**. Mer activation by rmGas6 and rmPros1.

in serum-free medium, using immunoprecipitation with antibodies to the individual receptors (or their C-terminal HA tag) followed by immunoblotting with an anti-phosphotyrosine antibody (see 'Materials and methods'). We found that both purified human and recombinant mouse Pros1 (hPros1 and rmPros1, respectively) were capable of rapidly activating Tyro3 autophosphorylation (**Figure 2A–C**). In

our hands, purified hPros1 was more stable biochemically and often more potent than rmPros1 (*Figure 2A*), and we therefore used the human protein for the majority of the Pros1 experiments described below. However, both the mouse and human ligands were active (*Figure 2B,C*). Consistent with these observations, both mouse and human Pros1 were also recently found to be active as ligands for a chimeric receptor comprised of the mouse Tyro3 ectodomain linked to the cytoplasmic domain of the R1 chain of the interferon (IFN)-gamma (γ) receptor (*Tsou et al., 2014*). (Activation of this chimera, expressed in CHO cells, was monitored by tyrosine phosphorylation of STAT1.)

Mouse Tyro3 was also strongly activated by rmGas6 (*Figure 2B–D*). In contrast, we found that mouse Axl could only be activated by rmGas6 (*Figure 2D–F*). While we detected 10 min activation of Axl at rmGas6 concentrations as low as 1 nM (*Figure 2D*), we could not detect Axl activation by either mouse Pros1 (*Figure 2E*) or human Pros1 (*Figure 2F*) at any concentration tested. Correspondingly, hGas6 but not mPros1 has been found to induce STAT1 phosphorylation downstream of a chimeric mAxl/γR1 receptor expressed in CHO cells (*Tsou et al., 2014*).

We used similar dose–response titrations to assess the ability of Gas6 and Pros1 to activate Mer. We found that both ligands were capable of activating this receptor in Mer-expressing MEFs, but that rmGas6 was more potent than hPros1 in this assay (*Figure 2G*). Finally, as our Mer-expressing MEFs have much lower receptor levels than Tyro3 or Axl-expressing MEFs, we also performed the same analysis using dexamethasone-treated mouse bone marrow-derived macrophages (Dex_Mφ). These cells have levels of Mer comparable to those of Tyro3 or Axl in the MEF cell lines and do not express any other TAM receptor (*McColl et al., 2009*; *Zagórska et al., 2014*; and data not shown). We found that both rmGas6 and hPros1/rmPros1 also function as a ligand for Mer in these cells (*Figure 2H* and *Figure 2—figure supplement 1*).

## The Gas6 Gla domain is essential for optimal TAM receptor activation

Together with several components of the blood coagulation cascade, including prothrombin, Protein C, and factors VII, IX, and X, both Gas6 and Pros1 carry 'Gla domains'—polypeptide segments of ~45 amino acids that contain clusters of 10–12 glutamic acid residues whose gamma carbons are post-translationally carboxylated in a vitamin K-dependent reaction (*Stitt et al., 1995*; *Nelsestuen et al., 2000*; *Huang et al., 2003*; *Bandyopadhyay, 2008*). This γ-carboxylation of glutamic acid is required for the Ca$^{2+}$-dependent binding of Gla domains to PtdSer, which is displayed on the surface of apoptotic cells (ACs) and enveloped viruses (*Bhattacharyya et al., 2013*). It is indispensible for the bioactivity of the Gla domain-containing proteins of the coagulation cascade and is a principal reason that vitamin K is an essential vitamin (*Freedman et al., 1995*; *Ishimoto et al., 2000*; *Bandyopadhyay, 2008*; *Rajotte et al., 2008*; *Lemke, 2013*). The anticoagulant warfarin and related blood thinners antagonize vitamin K-dependent γ-carboxylation of Gla domains (*Stafford, 2005*).

We first compared the ability of full-length vs Gla-less mouse Gas6 to activate Tyro3. Gla-less Gas6 is deleted for the first 115 residues of the protein, including the Gla domain, but retains all four EGF-like domains and the SHBG-like domain. We found that although both full-length and Gla-less rmGas6 were capable of activating Tyro3, the full-length protein was markedly (~20-fold) more potent (*Figure 3A*). When we performed this same comparison for Axl-expressing MEFs, we observed an even stronger dependence. Full-length Gas6 was a very potent Axl ligand (*Figure 3B*, left 6 lanes), but the Gla-less version was incapable of activating Axl at any concentration tested (up to 750 nM; *Figure 3B*, right 6 lanes). We found that this stark disparity was unique to Axl. When we compared the ability of full-length and Gla-less rmGas6 to activate Mer in Mer-expressing Dex-Mφ, we observed a pattern similar to that seen for Tyro3. Although both forms of the ligand were capable of activating Mer, full-length Gas6 was ~20-fold more potent than its Gla-less counterpart (*Figure 3C*). Thus, the TAM receptors fall into two groups based on features of their ligand and PtdSer activation profiles: Tyro3 and Mer are activated by both Gas6 and Pros1, whereas Axl is activated only by Gas6; and while Gla-less Gas6 is a substantially weaker ligand for Tyro3 and Mer, it is effectively dead as a ligand for Axl.

The difference in the ability of Gla-less Gas6 to activate Axl vs Tyro3/Mer is unrelated to any difference in the intrinsic tyrosine kinase activity of the three receptors. We found that a Tyro3–Axl chimeric receptor composed of the Tyro3 extracellular domain linked to the Axl tyrosine kinase, when expressed in TAM TKO MEFs, displayed diminished activation by Gla-less Gas6 similar to that seen with Tyro3 (*Figure 3A,D*). This suggests that the interaction of Gla-less Gas6 to Tyro3 and Mer

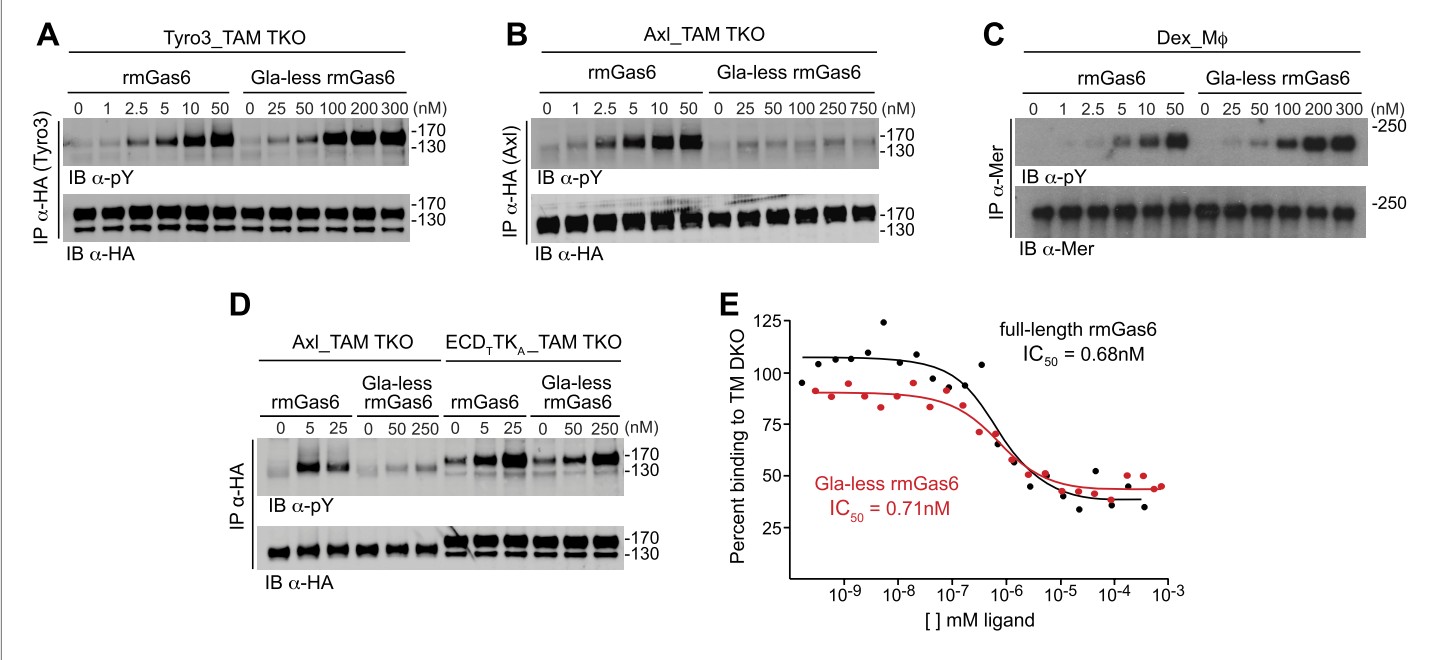

**Figure 3**. The role of the Gla domain in TAM receptor activation. (**A**, **B**, **C**) Tyro3-expressing MEFs, Axl-expressing MEFs, or dexamethasone-treated BM-derived macrophages expressing Mer, respectively, were stimulated with the indicated increasing concentrations of full-length rmGas6 or Gla-less rmGas6, respectively. Total cell lysates were immunoprecipitated with either HA antibodies (**A** and **B**) or Mer-specific antibodies (**C**) and subsequently subjected to SDS-PAGE and quantitative Licor western blotting (panel **C**, ECL western blotting) with the indicated antibodies. (**D**) MEFs expressing Axl or a chimeric receptor composed of the Tyro3 extracellular domain linked to the Axl tyrosine kinase domain (ECD$_T$TK$_A$) were stimulated with either full-length rmGas6 or Gla-less Gas6. Receptors were immunoprecipitated from cell lysates using an HA antibody, and immunoprecipitates were subjected to SDS-PAGE and Licor western blotting with anti-phosphotyrosine and HA antibodies. The higher basal activation of the ECD$_T$TK$_A$ construct may reflect its higher level of expression. (**E**) Binding assays were performed on Axl-expressing MEFs using a single concentration of $^{125}$I-labeled full-length rmGas6 in the presence of increasing concentrations of either unlabeled full-length rmGas6 (black) or Gla-less rmGas6 (red). Measured concentrations of unlabeled ligand required for 50% inhibition of displaceable binding (IC$_{50}$) are indicated.

is distinct structurally from its interaction with Axl. Together, these results indicate that the Gas6 Gla domain is required for optimal activation of Tyro3 and Mer, and that this requirement is absolute for Axl.

## The Gas6 Gla domain is dispensable for Axl binding

We asked whether the marked difference in Axl activation displayed by full-length vs Gla-less rmGas6 was reflected in a difference in Axl binding. Since there have been conflicting reports as to the relative binding affinity of full-length vs Gla-less Gas6 to Axl-Fc fusion proteins, as measured by using BIAcore sensor chips (*Nakano et al., 1997*; *Tanabe et al., 1997*; *Demarest et al., 2013*), we measured their Axl binding affinities in a more biologically relevant context; that is, to intact Axl receptor expressed on the surface of TM DKO MEFs (prepared from *Tyro3$^{-/-}$Mertk$^{-/-}$* embryos) in culture. We radio-iodinated full-length rmGas6 and then performed conventional competitive binding assays using displacement with increasing concentrations of either unlabeled full-length Gas6 or unlabeled Gla-less Gas6 (see 'Materials and methods').

We found that full-length and Gla-less Gas6 displayed essentially the same IC$_{50}$ for Axl binding—approximately 0.7 nM in this assay (*Figure 3E*). Thus, the inability of Gla-less Gas6 to activate Axl is unrelated to its ability to bind the receptor; and conversely, the ability of a Gas6 variant to bind Axl is unrelated to its ability to activate the receptor. These properties suggest that a crystal structure of the Axl ectodomain bound to Gla-less Gas6 (*Sasaki et al., 2006*) represents a ligand-occupied but inactive Axl configuration. In related work, we have found that these properties are also exceptionally important to Gas6–Axl binding and signaling in tissues in vivo, where Gas6 appears to be specifically and constitutively bound to Axl without significant activation of the receptor (*Zagórska et al., 2014*).

Consistent with the failure of Pros1 to activate Axl in culture (*Figure 2E,F*), we detected no binding of Pros1 to Axl-expressing MEFs (data not shown).

## Phosphatidylserine is present in MEF cultures

The importance of PtdSer for TAM ligand activity suggests that this phospholipid must be available to the purified TAM ligands that are added to MEF cultures (as in *Figures 2 and 3*). This is in spite of the fact that PtdSer is largely confined to the inner leaflet of the plasma membrane bilayer of non-apoptotic cells through the action of a set of $P_4$-ATPases—so-called flippases (*van Meer et al., 2008*). Consistent with this hypothesis, we found that the PtdSer-binding protein Annexin A5 antagonized Gas6 activation of Tyro3 in Tyro3_TAM TKO MEFs (*Figure 4A*). We therefore probed for the presence of PtdSer in MEF cultures directly. We co-incubated MEFs (grown under the same conditions we used for activation studies) with propidium iodide (PI) and the fluorescent Annexin B12 derivative

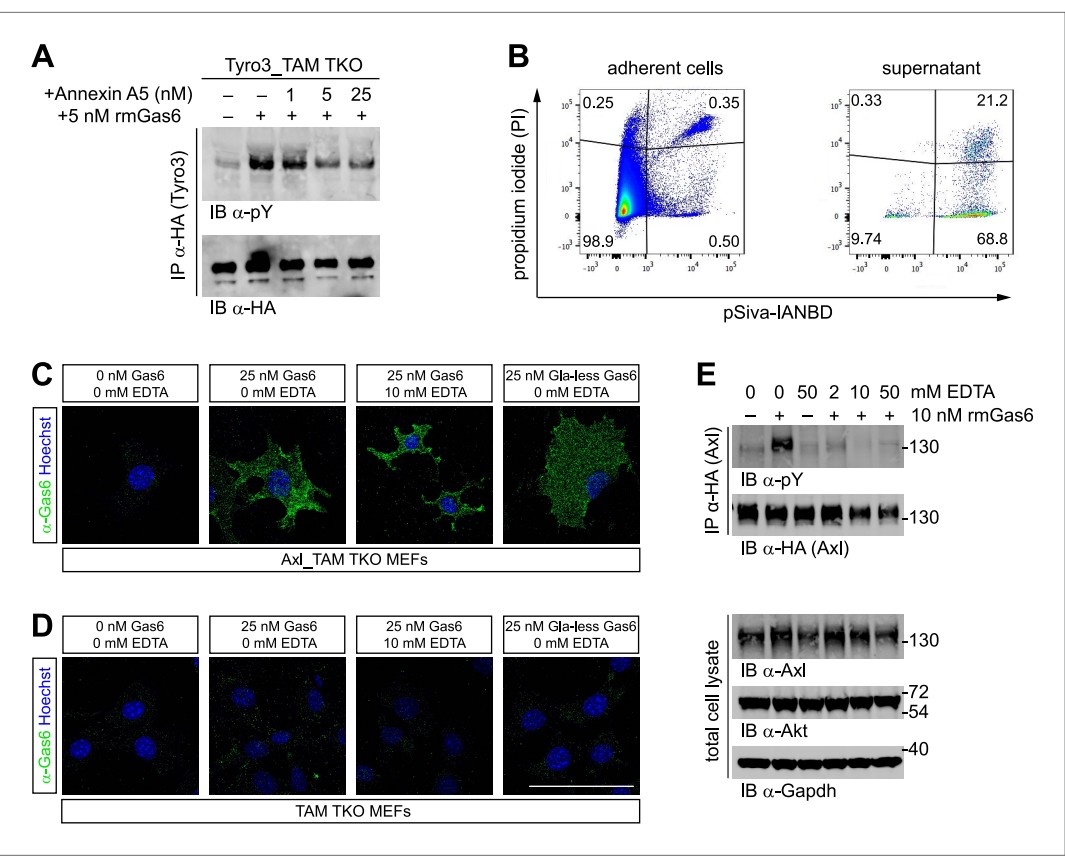

**Figure 4**. Phosphatidylserine and TAM activation. (**A**) Tyro3-expressing MEFs were treated with the indicated concentrations of Annexin A5 for 10 min and then stimulated with 5 nM full-length rmGas6 for 10 min. Total cell lysates were immunoprecipitated with HA antibodies and subsequently subjected to SDS-PAGE and western blotting with the indicated antibodies. (**B**) FACS analysis. Adherent cells (left) and cells from culture medium ('supernatant', right) from Axl_TAM TKO MEF cultures were analyzed by staining with propidium iodide, which is only taken up by dead cells, and pSIVA, which only fluoresces when bound to PtdSer (right panels). Numbers in the four quadrants of the panels indicate percent of signal in that quadrant. 21.2% of the gated material in the MEF culture supernatant (upper right quadrant in the right panel) represents PtdSer-expressing apoptotic cells. (**C** and **D**) Axl_TAM TKO MEFs (**C**) and control TAM TKO MEFs (**D**) were incubated +/− rmGas6 or Gla-less rmGas6 (as indicated) and +/− 10 mM EDTA for 90 min at 4°C and then live-stained with an anti-Gas6 antibody (green) and Hoechst to visualize nuclei. (**E**) Axl-expressing Axl_TAM TKO MEFs were stimulated +/− 10 nM rmGas6 in the presence of the indicated concentrations of EDTA for 10 min. Total cell lysates were either: (top two blots) immunoprecipitated with HA antibodies and subsequently subjected to SDS-PAGE and western blotting with the indicated antibodies; or (bottom three loading control blots) western blotted with the indicated antibodies. Scale bar (for **C** and **D**) 50 μm.

pSIVA, a polarity-sensitive biosensor that fluoresces only when bound to PtdSer (*Kim et al., 2010*; *Ruggiero et al., 2012*), and then analyzed the cells by flow cytometry. We detected significant number of pSIVA+PI− apoptotic cells and pSIVA+PI+ late apoptotic/necrotic cells in the culture medium and even a measurable number of pSIVA+PI− and pSIVA+PI+ cells in the adherent MEFs on the plate (*Figure 4B*). Together, these results demonstrate that there are PtdSer-displaying membranes present in the MEF cultures. It is important to note that the molecular weight of PtdSer is ~385 g/mol, while Gas6 and Pros1 have molecular weights of ~80,000 g/mol, and that the crystal structure of the pro-thrombin Gla domain bound to lyso-PtdSer contains only a single molecule of this phospholipid (*Huang et al., 2003*). Thus, a relatively low level of PtdSer can easily be in molar excess over the Gla domain-containing proteins with which it interacts. In this regard, the addition of either PtdSer-displaying enveloped viruses (*Bhattacharyya et al., 2013*) or PtdSer-displaying apoptotic cells (*Zagórska et al., 2014*) has been found to markedly shift the dose–response curves for Pros1 activation of Mer and Gas6 activation of Axl/Mer to lower ligand concentrations.

## Gas6 binding to Axl is calcium independent but Gas6 activation of Axl is calcium dependent

The binding of Gla domains to PtdSer is also $Ca^{2+}$-dependent (*Huang et al., 2003*) and can be rapidly disrupted by $Ca^{2+}$-chelating agents such as EDTA. We therefore used immunocytochemistry to ask if rmGas6 binding to Axl in Axl-expressing MEFs is $Ca^{2+}$-dependent. We found that it is not: binding of rmGas6 (at 25 nM) to the surface of Axl-expressing MEFs was readily observed with an anti-Gas6 antibody at 90 min after application (at 4°C), and this binding was unaffected by the inclusion of 10 mM EDTA (*Figure 4C*). We also detected equivalent binding of Gla-less Gas6 (*Figure 4C*). In control experiments, no Gas6 binding was detected in TAM TKO MEFs (*Figure 4D*). These results argue that the observed Gas6 binding is via SHBG domain binding to Axl and not via Gla domain binding to any PtdSer expressed by the MEFs. We then asked whether Gas6 bound to Axl in the presence of EDTA (*Figure 4C*, third panel) is capable of activating Axl. We found that it is not: although 10 nM rmGas6 resulted in strong Axl activation in the absence of EDTA, the inclusion of 2, 10, or 50 mM EDTA all blocked this activation (*Figure 4E*). These results indicate that the binding of full-length Gas6 alone, in the absence of simultaneous $Ca^{2+}$-dependent binding of its Gla domain to PtdSer, is insufficient to trigger Axl activation.

## Gas6 activation of Axl in vivo

We next extended these cell culture observations to an in vivo setting. We injected either full-length or Gla-less rmGas6 intravenously (IV) into *Gas6*−/− mice and then used immunohistochemistry to monitor the binding of these injected proteins to splenic red pulp macrophages. F4/80+ red pulp macrophages express both Axl and Mer and are normally also Gas6+ (*Zagórska et al., 2014*). By 30 min after IV injection, we observed equally strong binding of both full-length and Gla-less rmGas6 to these cells (*Figure 5A*). However, this equivalent in vivo binding was not reflected in equivalent Axl activation. When we immunoprecipitated Axl from the spleen 30 min after IV injection of full-length Gas6, we observed that Axl was strongly activated (*Figure 5B*). In marked contrast, Axl immunoprecipitated after injection of Gla-less rmGas6 showed zero activation (*Figure 5B*). This was in spite of the fact that equivalent amounts of full-length and Gla-less Gas6 could be recovered from the spleen (*Figure 5B*). The PtdSer required for activation of Axl by full-length rmGas6 injected IV is presumably provided by both apoptotic cells and circulating PtdSer-exposing microparticles, derived from platelets, erythrocytes, leukocytes, and endothelial cells, which are abundant in the blood (*Lacroix and Dignat-George, 2012*).

## TAM signaling differences reside in the Ig-like domains of the receptors

We found that the distinct receptor–ligand activation profiles described above reflect differences that are intrinsic to the Ig-like domains of the TAM ectodomains. We compared the ability of rmGas6 and hPros1 to activate TAM receptor autophosphorylation in MEF lines expressing either full-length Tyro3 or a chimeric receptor composed of the Tyro3 extracellular domain linked to the tyrosine kinase domain of Axl (*Figure 6A*). This chimeric receptor was also expressed well in clonal TAM TKO MEF lines (*Figure 6B* and *Figure 6—figure supplement 1*). The activation profiles of both rmGas6 (*Figure 6C*) and hPros1 (*Figure 6D*) were similar for MEF lines expressing equivalent levels of wild-type Tyro3 or the chimeric Tyro3–Axl hybrid. We found that specificity resides within the two N-terminal Ig-like domains of the receptors, and that both of these domains are required: a hybrid receptor

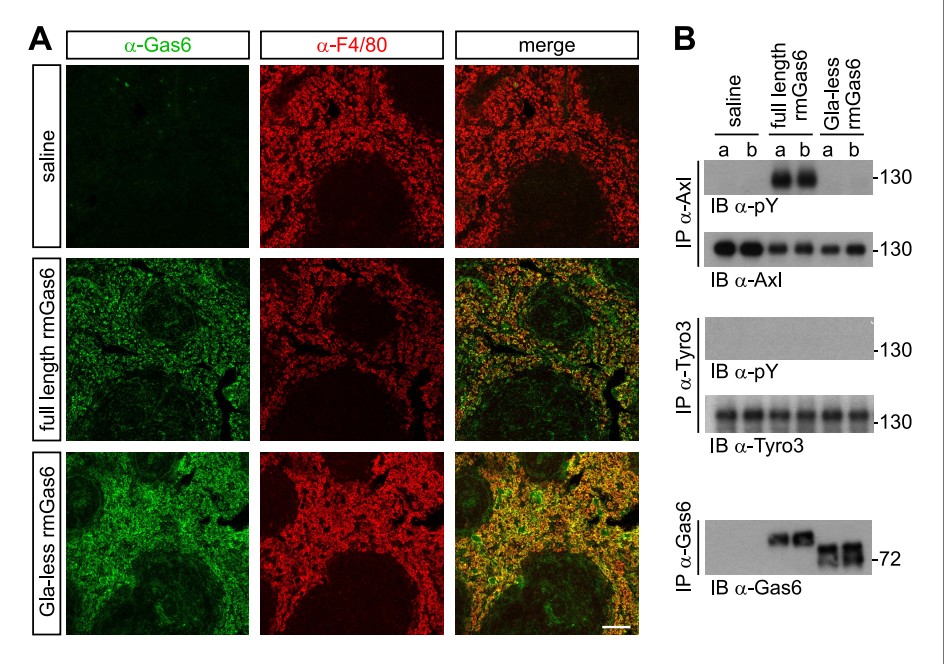

**Figure 5**. Gla-less Gas6 binds but does not activate Axl in vivo. (**A**) Sections of spleens from *Gas6⁻/⁻* mice 30 min after IV injection of saline (top row), 30 μg full-length rmGas6 (middle row), or 30 μg Gla-less rmGas6 (bottom row), and stained with an anti-Gas6 antibody (first column, green) and an anti-F4/80 antibody to identify splenic red pulp macrophages (second column, red). Merged images from the first and second columns are displayed in the third column. (**B**) Splenic lysates from *Gas6⁻/⁻* mice injected IV as in (**A**) were immunoprecipitated with the indicated antibodies 30 min after injection, and the immunoprecipitates then immunoblotted for pY and Axl (top two panels), pY and Tyro3 (middle two panels), or Gas6 (bottom panel).

composed of the two Ig-like domains of Tyro3 linked to the remaining domains of Axl (*Figure 6A*) was, like Tyro3, potently activated by Pros1 (*Figure 6D*, lanes 5–8), whereas a chimera that contained only the first Ig-like domain of Tyro3 (*Figure 6A*) was refractory to activation by Pros1 (*Figure 6D*, lanes 1–4). These results indicate that the intrinsic catalytic activity of Axl and Tyro3 are stimulated to comparable levels by ligand binding—of either Gas6 or Pros1–to the Tyro3 extracellular domain.

## Genetic delineation of TAM receptor–ligand interactions in the retina

We extended our in vitro observations of differential TAM interactions to two in vivo settings in which genetic analyses have shown that TAM signaling plays an essential role. The first of these is the phagocytosis of the distal tips of photoreceptor's (PR) outer segments by the retinal pigment epithelial (RPE) cells of the eye (*Prasad et al., 2006*; *Burstyn-Cohen et al., 2012*). As noted above, Mer is absolutely required for this phagocytosis (*D'Cruz et al., 2000*; *Duncan et al., 2003*; *Prasad et al., 2006*; *Nandrot and Dufour, 2010*; *Ostergaard et al., 2011*). In *Mertk⁻/⁻* mice, a phagocytic deficiency results in the nearly complete death of PRs by 12 weeks of age (*Prasad et al., 2006*; *Burstyn-Cohen et al., 2012*). We have previously shown that *Tyro3* mouse mutants, *Gas6* mouse mutants, and retina-specific *Pros1* mouse mutants all have a normal number of PRs at this time (*Prasad et al., 2006*; *Burstyn-Cohen et al., 2012*); but that *Gas6/Pros1* double mutants display PR death and retinal degeneration that fully phenocopies the degeneration of the *Mertk* mutants (*Burstyn-Cohen et al., 2012*).

These analyses demonstrate that Pros1 acts as an effective Mer activator during RPE cell phagocytosis but not that this occurs through direct binding of Pros1 to Mer. To address this question, we generated *Tyro3⁻/⁻Gas6⁻/⁻* double mutants. In these mice, the only TAM ligand remaining is Pros1, and the only TAM receptor through which it can bind and signal in RPE cells is Mer. As noted above, RPE cells do not express Axl (*Burstyn-Cohen et al., 2012*), which, in addition, is only activated by Gas6 (*Figure 2E,F*). As described previously, PR death can be assessed quantitatively in retinal sections by measuring the radial thickness of the outer nuclear layer (ONL), which is composed exclusively of

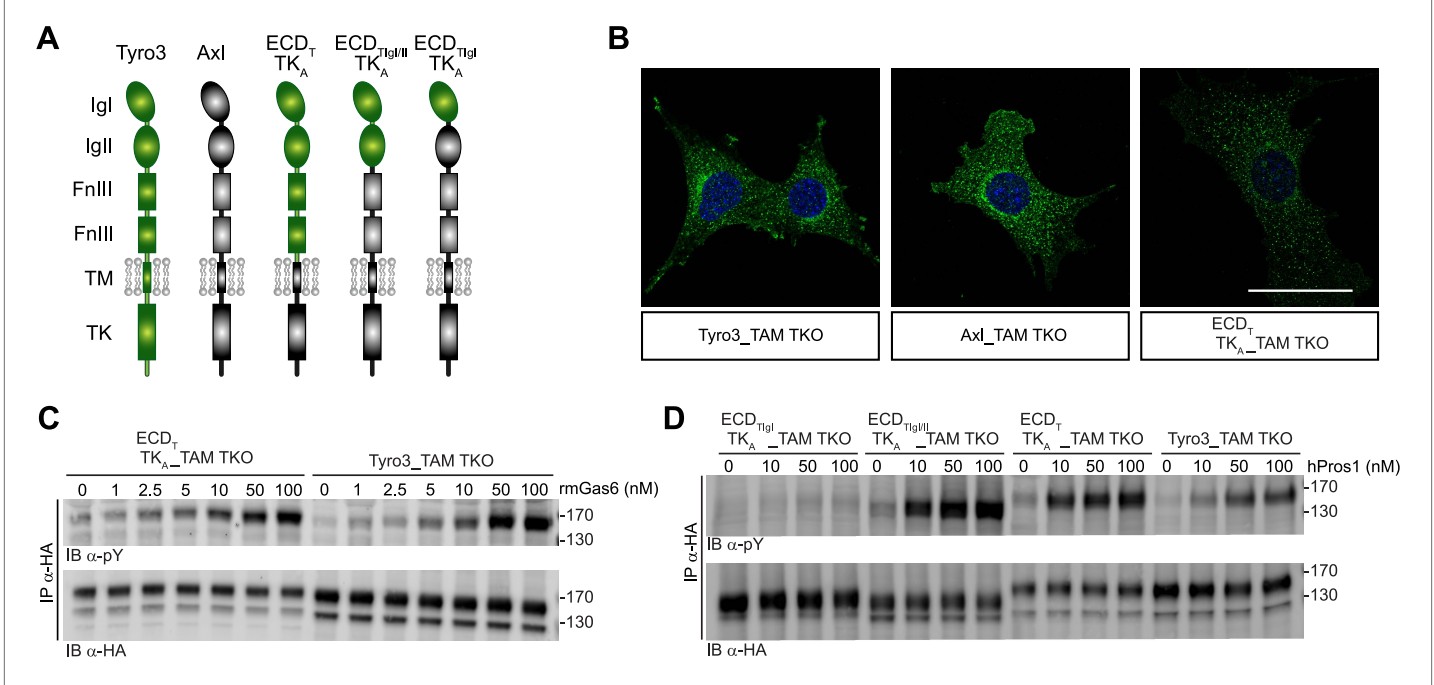

**Figure 6**. Differences in TAM receptor activation are not due to differences in TAM kinase activity. (**A**) Schematic of wild-type Tyro3 and Axl, a Tyro3/Axl chimeric receptor carrying the complete Tyro3 ectodomain linked to the Axl transmembrane (TM) and tyrosine kinase (TK) domains (ECD$_T$TK$_A$), and chimeric receptors carrying both Tyro3 Ig domains (ECD$_{TIgI/II}$TK$_A$) or only the first Tyro3 Ig domain (ECD$_{TIgI}$TK$_A$). FnIII: fibronectin type III repeat. (**B**) Immunostaining of Tyro3-, Axl-, and Tyro3/Axl chimera expressing TAM TKO MEFs, using an anti-HA antibody (green). Bar: 50 μm. (**C** and **D**) MEFs expressing Tyro3 or the indicated Tyro3/Axl chimeric receptors were stimulated with increasing concentrations of either rmGas6 or hPros1, respectively. Cell lysates were immunoprecipitated with anti-HA antibodies and subjected to SDS-PAGE electrophoresis followed by quantitative Licor western blotting with the indicated antibodies.

The following figure supplement is available for figure 6:

**Figure supplement 1**. Surface expression of Tyro3/Axl chimeric receptors.

PR nuclei (**Burstyn-Cohen et al., 2012**). In 10–12-week old wild-type (**Figure 7A**, first panel), *Gas6*$^{-/-}$ (**Figure 7A**, third panel), and *Tyro3*$^{-/-}$ retinae (**Figure 7A**, fourth panel), the ONL has a normal radial thickness of ~50 μm and consists of 12–15 compact PR nuclei (**Prasad et al., 2006**; **Burstyn-Cohen et al., 2012**). In marked contrast, the *Mertk*$^{-/-}$ ONL is reduced to a thickness of only 1–4 PR nuclei (**Figure 7A**, second panel) (**Burstyn-Cohen et al., 2012**). (The **Figure 7A** panel illustrates a region of severe PR degeneration in which the ONL is only one nucleus thick.) We found that the *Tyro3*$^{-/-}$*Gas6*$^{-/-}$ ONL is also of wild-type thickness (**Figure 7A**, fifth panel). Thus, Pros1–Mer signaling alone is sufficient to mediate apparently normal phagocytosis in the retina.

We also assessed the ability of hPros1 to activate Mer in a dissected RPE layer preparation in vitro (**Prasad et al., 2006**; **Burstyn-Cohen et al., 2012**). As before (**Prasad et al., 2006**), we isolated the eyecup from either wild-type or *Tyro3*$^{-/-}$ adult mice and removed the cornea, lens, and retina to leave the apical surface of the RPE cell layer exposed. We then treated this surface, where Mer and Tyro3 are localized (**Prasad et al., 2006**), with hPros1. We have shown previously that purified Pros1 can enhance Mer autophosphorylation in this preparation (**Prasad et al., 2006**; **Burstyn-Cohen et al., 2012**). As shown in **Figure 7B** (upper two panels), we demonstrate that purified Pros1 can also activate Tyro3 in this preparation and can activate Mer in both wild-type and *Tyro3*$^{-/-}$ RPE cells (**Figure 7B**, lower two panels). Thus, Pros1 induces Mer autophosphorylation in the RPE directly, in the absence of any other TAM receptor, and this Pros1-to-Mer activation alone is sufficient to maintain a normal number of PRs in the retina in vivo.

## Genetic delineation of TAM receptor–ligand interactions in the testes

A second setting in which PtdSer-dependent TAM signaling plays an essential role is the homeostatic phagocytosis of apoptotic cells in the seminiferous tubules of the testes (**Lu et al., 1999**). This setting

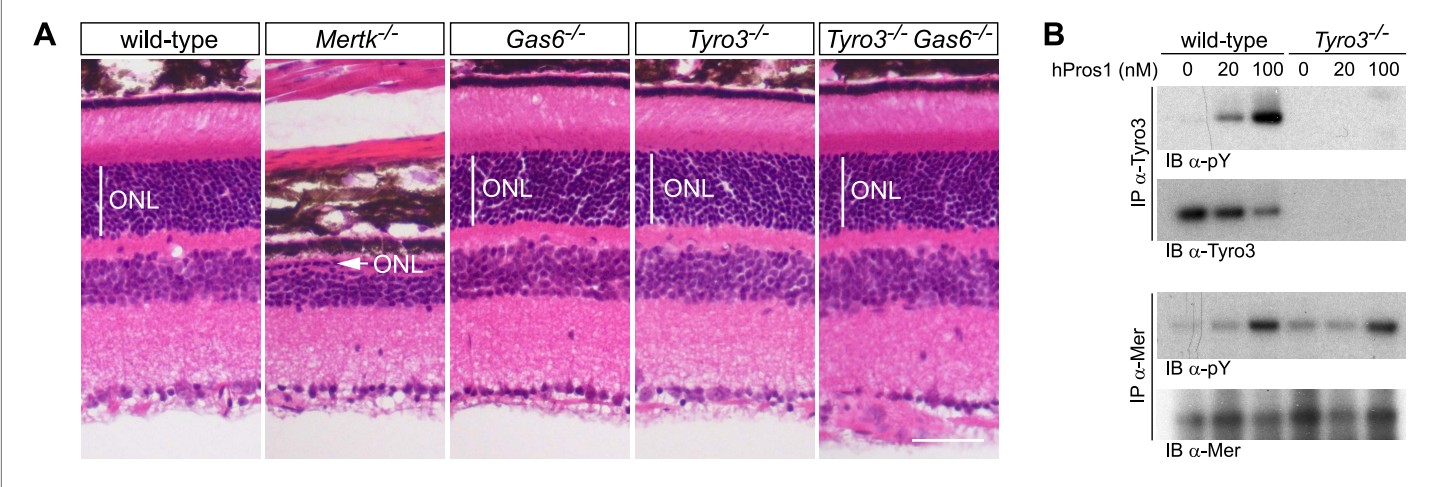

**Figure 7**. The Pros1–Mer signaling axis is sufficient for RPE phagocytosis in the retina. (**A**) Dorsal–ventral H&E stained sections from 12–14 week mouse retinae. The outer nuclear layer (ONL), composed exclusively of PR nuclei, is delimited by the vertical white line. The ONL in the *Mertk*[−/−] mutant (second panel) is reduced to a thickness of a single nucleus (arrow) and the outer segment (OS) layer above is absent. In contrast, the ONL in the *Gas6*[−/−], *Tyro3*[−/−], and the *Tyro3*[−/−]*Gas6*[−/−] retinae is of a thickness that is indistinguishable from wild-type control. Bar: 50 μm. (**B**) Pros1-mediated activation of Mer in the RPE cell layer is independent of Tyro3 expression. Eye cups were acutely isolated from wild-type and *Tyro3*[−/−] mice. Cornea, iris, and lens were removed, leaving the RPE cell layer exposed. The eyecup was cultured under starvation conditions for 3 hr, and subsequently stimulated with increasing concentrations of hPros1. Cell lysates were immunoprecipitated with anti-Tyro3 or anti-Mer antibodies and receptor activation monitored by immunoblotting with anti-phosphotyrosine antibody (pY).

is distinct from retina, where RPE cells phagocytize only a portion of a living cell. In the testes, Sertoli cells must engulf and clear millions of apoptotic germ cells that are generated normally during every cycle of mammalian spermatogenesis (>10[8]/day in a human male). Sertoli cells express all three TAM receptors and both ligands (*Lu et al., 1999*; *Chen et al., 2009*), and *Tyro3/Axl/Mer* triple mouse mutants are infertile due to the toxic accumulation of ACs (*Lu et al., 1999*). We used immunostaining with antibodies to activated (cleaved) caspase 3 (cCasp3; see 'Materials and methods') to quantify the number of ACs present in the 10–12-week seminiferous tubules across an informative set of TAM receptor and ligand mutants. We counted all activated cCasp3[+] cells throughout an entire testis section and then normalized these counts to the number of tubule cross-sections present per testis section, which averaged 262.1 ± 20.6 (1 S.D.) and did not vary significantly for any of the genotypes we analyzed (data not shown).

We found that the major receptor required for Sertoli cell phagocytosis of apoptotic germ cells in the mouse testis is Mer (*Figure 8A,B* and *Figure 8—figure supplement 1*). While *Tyro3*[−/−], *Axl*[−/−], and *Gas6*[−/−] single mutant mice all displayed wild-type number of cCasp3[+] ACs—~0.02 cells per tubule cross-section (*Figure 8A*, and *Figure 8—figure supplement 1*)–*Mertk*[−/−] mice displayed nearly 20 times more (*Figure 8B*). In contrast to other tissues we have analyzed (e.g., the spleen), the number of ACs was not increased in *Mertk*[−/−]*Axl*[−/−] double knock-outs compared to *Mertk*[−/−] single knock-outs (*Figure 8—figure supplement 1A,C*). More tellingly, *Tyro3*[−/−]*Axl*[−/−] double knock-outs displayed wild-type number of ACs in the testes (*Figure 8—figure supplement 1A,C*), and this was also the case for *Tyro3*[−/−]*Gas6*[−/−] double knock-outs, in which the only possible TAM signaling axis is Pros1 to Mer (*Figure 8A,B*).

In contrast to the retina, we observed a significant effect of *Tyro3* mutation in the presence of an existing *Mertk* mutation: *Tyro3*[−/−]*Mertk*[−/−] double mutants displayed substantially more ACs in the testis (~2 cCasp3[+] cells per each tubule cross-section) than did *Mertk*[−/−] single mutants (*Figure 8—figure supplement 1A,C*). Combining all three receptor knock-outs in *Tyro3*[−/−]*Axl*[−/−]*Mertk*[−/−] triple mutants led to a massive accumulation of ACs and cCasp3[+] material that made accurate counting of ACs impossible (*Figure 8—figure supplement 1A*). This suggests that in the absence of Mer and Tyro3, Axl can also mediate AC phagocytosis. Finally, we examined the effect of removing all Gas6 and half the normal Pros1 throughout the mouse and found that the *Pros*[fl/−]*Gas6*[−/−] testes also contained

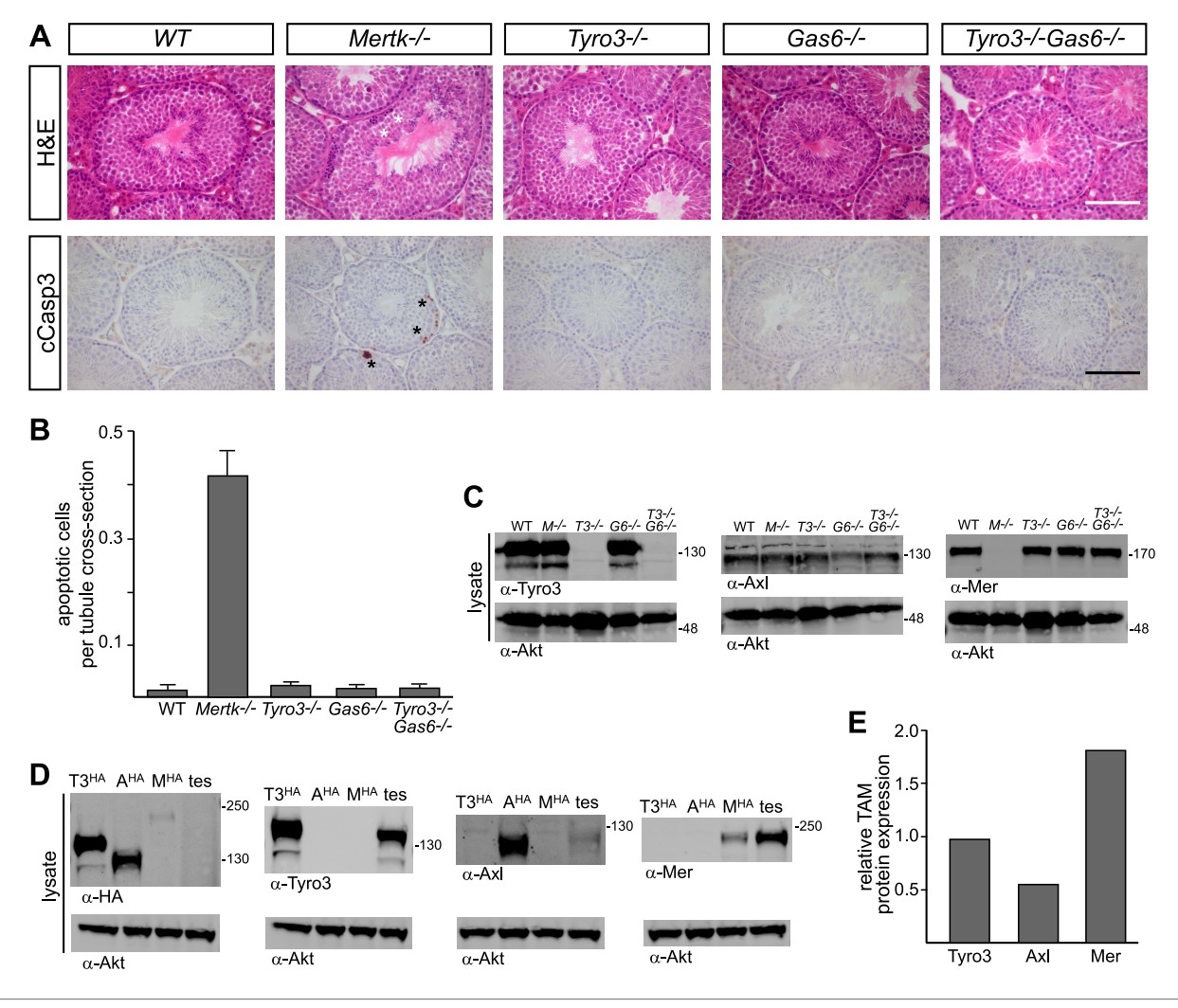

**Figure 8**. Pros1–Mer signaling is sufficient for TAM-dependent homeostatic clearance of apoptotic cells in the testis. (**A**) Representative H&E images (top row) or immunostaining for cleaved caspase 3 (cCasp3) (bottom row) in testis sections from 12–14 week old wild-type and TAM ligand and/or TAM receptor mutants. For the *Mertk*−/− sections (second column), apoptotic cells are highlighted with white asterisks in the H&E-stained section, and cCasp3+ cells are highlighted with black asterisks. Bar: 100 μm. (**B**) Quantitative analysis of the number of apoptotic cells in wild-type and various TAM receptor/ligand mutants. Cleaved caspase 3+ cells and tubule cross-sections were counted in four testis cross-sections per mouse. Data are expressed as average number of ACs per number of total tubule cross-sections in each testis section. Error bars represent standard error of the mean for three independent animals. (**C**) TAM receptor expression in TAM receptor and TAM ligand mouse mutants. Testes were collected from 12- to 14-week old mice of the indicated genotypes and lysed. Lysates were subjected to SDS-PAGE and immunoblotting with anti-Tyro3, Axl, and Mer antibodies, with anti-Akt serving as a loading control. (**D**) Comparative analysis of TAM expression in the testis. Lysates from either wild-type testis or TAM TKO MEFs expressing HA-tagged Tyro3, Axl, or Mer were subjected to SDS-PAGE and subsequent quantitative Licor immunoblotting with the indicated antibodies. (**E**) Band intensity of the HA-tagged TAM receptors relative to each other and to wild-type testis was calculated using Licor Odyssey software and utilized to calculate the TAM protein expression relative to Tyro3 in the wild-type testis.

The following figure supplement is available for figure 8:

**Figure supplement 1**. Tyro3 and Mer are key regulators of homeostatic apoptotic cell clearance in the testis.

a wild-type number of ACs (***Figure 8—figure supplement 1C***). Thus in both the testes and the retina (***Burstyn-Cohen et al., 2012***), only half the normal level of only a single TAM ligand (Pros1) is sufficient to maintain an essential level of TAM-dependent phagocytosis.

All three TAM receptors are detectable by western blot in the testis (***Figure 8C***), and there was no significant compensatory change in Tyro3, Axl, or Mer expression in *Mertk*−/−, *Tyro3*−/−, *Gas6*−/−, or

*Tyro3⁻/⁻Gas6⁻/⁻* mice (*Figure 8C*). We used comparative signal intensities on Licor immunoblots of lysates from testis and TAM TKO MEFs expressing HA-tagged Tyro3, Axl, and Mer (*Figure 8D*) to determine that the relative protein expression level of TAM receptors in the testis is Mer > Tyro3 > Axl (*Figure 8E*). Finally, we confirmed that the phenotypes we analyzed at 10–12 weeks were fully developed by this time. We measured the number of cCasp3⁺ cells in wild-type vs *Tyro3⁻/⁻Mertk⁻/⁻* mice at 10–12 weeks vs 9–12 months and saw that this number was the same in both the young and aged testis (*Figure 8—figure supplement 1D*). Together, these results indicate that Mer is the major facilitator of steady-state homeostatic phagocytosis in both the retina and the testis, and that binding and activation of this receptor by Pros1 alone at only half its normal gene dose is sufficient to maintain normal levels of phagocytosis.

## Discussion

### TAM rules of engagement

From the above analyses, we draw seven salient conclusions with respect to TAM receptor–ligand–phospholipid interactions, which are summarized in *Figure 9*. First, Gas6 alone is capable of binding to and activating Tyro3, Axl, and Mer when these receptors are expressed in isolation and is especially potent as a ligand for Axl. Second, Pros1 alone is capable of binding to and activating Tyro3 and Mer but does not function as a ligand for Axl. Third, the PtdSer-binding Gla domain of Gas6, PtdSer itself, and calcium are all required for optimal receptor activation but none is required for receptor binding. Fourth, ligand binding does not translate into receptor activation: Gla-less Gas6 binds as well to Axl as does its full-length counterpart, but is dead as an activator, and full-length Gas6 binds to Axl in the presence of EDTA but does not activate. Fifth, receptor heterodimerization is not an essential feature of TAM activation, since the TAM receptors can be activated by purified single ligands when expressed as single receptors in MEFs in the absence of any other TAM receptor, and in the retina, Mer activation in wild-type and Tyro3-deficient RPE cells is equivalent. Sixth, in the two settings of PtdSer-dependent 'homeostatic' phagocytosis—by RPE cells in the adult retina and Sertoli cells in the

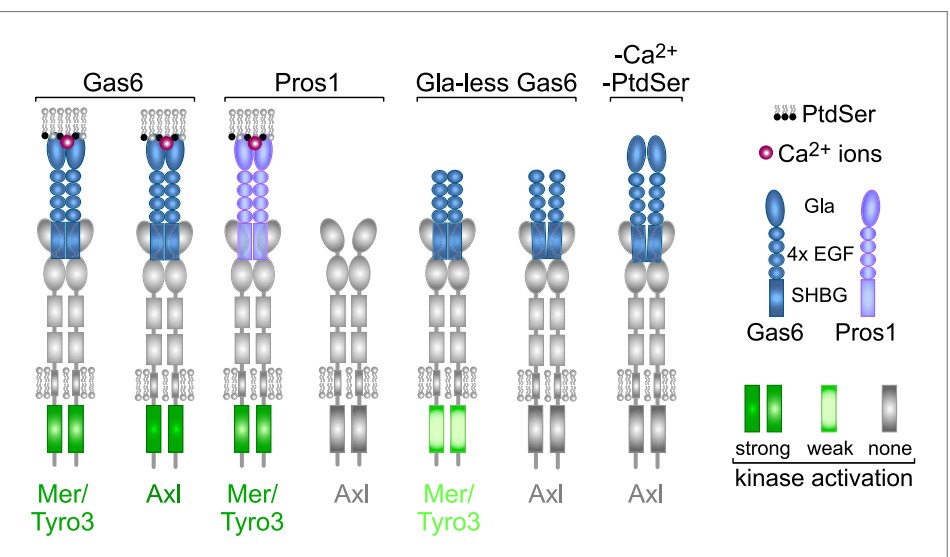

**Figure 9**. Rules of engagement for TAM receptor ligand interaction and signaling. Gas6 activates all three TAM receptors independently, but Axl is uniquely dependent on Gas6: Pros1 activates Tyro3 and Mer but not Axl. Optimal activation of any receptor by any ligand requires the simultaneous presence of PtdSer, which binds to the Gla domain of the ligands, and calcium ions (Ca²⁺). Gla-less Gas6 is dead as a ligand when assayed against Axl in isolation. Its Axl-bound orientation is schematized differently from Tyro3- and Mer-bound Gla-less Gas6, since the latter two result in partial kinase activation, although there are no structural data for these complexes. The right-most signaling configuration, in which full-length Gas6 is also inactive as an Axl ligand in the absence of PtdSer and Ca²⁺, is speculation based on the results of this paper and data in (*Zagórska et al., 2014*). See text for details.

adult testes—Mer is the functionally predominant TAM receptor. And seventh, in these settings, Pros1 binding to and activation of Mer alone is sufficient to ensure wild-type levels of phagocytosis.

Prior to this study, the relative binding and selectivity of Gas6 and Pros1 to the individual TAM receptors were incompletely understood. While Gas6 was widely regarded as a universal TAM ligand, the role of Pros1 in TAM-mediated signaling has, until recently, been controversial (*Godowski et al., 1995*; *Stitt et al., 1995*; *Burstyn-Cohen et al., 2012*), and older data regarding the relative potency of Gas6 and Pros1 in the binding and activation of the specific TAM receptors in vitro were conflicting (*Ohashi et al., 1995*; *Nagata et al., 1996*; *Chen et al., 1997*). Perhaps most importantly, all prior analyses of receptor activation following the addition of TAM ligands to cultured cells were confounded by the presence of endogenous TAM receptors in the cell lines used for these analyses. Together with earlier work from our group in the retina (*Burstyn-Cohen et al., 2012*), the studies of *Figures 7 and 8* represent the only use of molecular genetics to parse differential TAM receptor–ligand signaling interactions in vivo.

## A lipid-regulated, calcium-regulated kinase in which the lipid and calcium are both extracellular

Our experiments suggest a model in which the interaction between distinct TAM receptors and ligands defines a signaling signature that is specific for each ligand–receptor pair (*Figure 9*). This is best illustrated in the case of Gas6 activation of Axl, which is uniquely dependent on Gas6 for activation. Both full-length Gas6 and a Gla-less Gas6 variant that lacks its PtdSer-binding Gla domain bind to Axl with the same sub-nanomolar equilibrium dissociation constant, yet the former is an extremely potent Axl agonist while the latter is inactive, whether assayed in vitro or in vivo. Furthermore, full-length Gas6 cannot activate Axl in the presence of EDTA, which disrupts the binding of the Gas6 Gla domain to PtdSer. In a related study (*Zagórska et al., 2014*), we have found that just as Axl is uniquely dependent on Gas6 for activation, so is Gas6 uniquely dependent on Axl for its stable localization in several different tissues in vivo. We find that Axl is constitutively pre-bound to Gas6 in these tissues but that this binding does not result in significant Axl activation (*Zagórska et al., 2014*). Together, all of these observations suggest that the induced exposure of PtdSer is actually the regulated trigger for TAM activation and that the TAM receptor–ligand pair is in reality a PtdSer detector. They also suggest that a crystal structure of the two Ig-like domains of the Axl ectodomain bound to two molecules of Gla-less Gas6 (*Sasaki et al., 2006*) reflects the structure of an inactive complex.

We propose that an effective TAM signaling complex in vivo is always tripartite, in that it is always composed of a TAM receptor, a Gla domain-containing TAM ligand, and the phospholipid PtdSer (*Figure 9*). This configuration is unique to the TAMs. In this regard, it is important to note that all settings of TAM-dependent 'homeostatic phagocytosis' are strictly co-dependent upon the exposure of PtdSer on the surface of the phagocytic target (*Shiratsuchi et al., 1997*; *Ruggiero et al., 2012*; *Lemke, 2013*). In the case of RPE cell phagocytosis of juxtaposed PR outer segment tips, PtdSer is locally exposed only on these outer segment tips (i.e., not on the remainder of the photoreceptor) and only during the narrow window after subjective dawn in which phagocytosis occurs each day (*Ruggiero et al., 2012*). The activity of Gla domains is also $Ca^{2+}$-dependent, as these ions both stabilize the folded structure of the domain and also interact with PtdSer (*Bandyopadhyay, 2008*; *Huang et al., 2009*). Thus, the TAM receptor tyrosine kinases represent a novel addition to the set of enzymes, such as protein kinase C, whose activities are regulated by both calcium and lipid binding (*Lemmon, 2008*; *Leonard and Hurley, 2011*). The key difference for the TAMs is that neither the lipid nor the calcium binds the kinase directly. Instead, they bind the TAM ligands, which are outside the cell.

## Axl–Gas6 and inflammation

The dependence of Axl on Gas6 may be relevant to the fact that all of the very low level of Gas6 that is present in blood appears to be complexed with soluble Axl (sAxl) extracellular domain (*Ekman et al., 2010c*). Upon Axl activation, this 'ectodomain' is proteolytically cleaved from the rest of the receptor (*Costa et al., 1996*; *O'Bryan et al., 1995*; *Wilhelm et al., 2008*), which results in the generation of an sAxl–Gas6 complex. In related work, we have shown that this complex is also generated when Axl is activated in tissues subsequent to the injection of activating (cross-linking) α-Axl antibodies (*Zagórska et al., 2014*). Although there are conflicting reports as to the presence of soluble Tyro3 and Mer ectodomains in serum and the possibility that Gas6 might also be bound to soluble Mer

(*Sather et al., 2007*; *Ekman et al., 2010c*), antibody depletion of Gas6 from serum does not alter the gel filtration profile of either soluble Mer or Tyro3 (*Ekman et al., 2010c*).

Elevated blood levels of soluble Axl have recently been reported to mark a variety of human disease and trauma states, including aortic aneurysm (*Ekman et al., 2010b*), lupus flares (*Zhu et al., 2014*), pneumonia infection (*Ko et al., 2014*), preeclampsia (*Liu et al., 2014*), coronary bypass (*Lee et al., 2013*), obesity and insulin resistance (*Hsiao et al., 2013*), and limb ischemia (*Ekman et al., 2010a*). We suggest that the generation of a soluble Axl–Gas6 complex is triggered by the induced cellular exposure of PtdSer in many of these settings.

### Differential TAM activation in human disease

Aberrant TAM signaling in human disease, most notably in cancer, is now a major research focus (*Leconet et al., 2013*; *Meyer et al., 2013*; *Schlegel et al., 2013*), and small-molecule TAM tyrosine kinase inhibitors and antibodies that inhibit TAM ligand–receptor binding are in development as cancer therapies (*Holland et al., 2010*; *Ye et al., 2010*; *Brandao et al., 2011*; *Schlegel et al., 2013*; *Paccez et al., 2014*). These same inhibitors have been proposed for the treatment of enveloped virus infections (*Bhattacharyya et al., 2013*; *Shibata et al., 2014*). Conversely, TAM activators have been proposed as possible treatments for several autoimmune indications (*Rothlin and Lemke, 2010*; *van den Brand et al., 2013*).

In each of these settings, our results indicate that it will be critical to know which TAM receptor to target. This will be particularly important in the context of cancer, since tumors profiled in the TCGA database (https://tcga-data.nci.nih.gov/tcga/) that exhibit mutation or expression changes in the *Tyro3*, *Axl*, and *Mertk* genes display a strong tendency towards mutually exclusive TAM changes across tumor types. At the same time, our studies here, related work in the immune system (*Zagórska et al., 2014*), and prior analyses (*Scott et al., 2001*; *Rothlin et al., 2007*; *Seitz et al., 2007*; *Burstyn-Cohen et al., 2009*, *2012*; *Carrera Silva et al., 2013*) all demonstrate that distinct normal functions are preferentially assumed by individual TAM receptors and ligands in macrophages, dendritic cells, Sertoli cells, endothelial cells, and RPE cells. Our delineation of the rules of engagement for TAM signaling (*Figure 9*) therefore has important implications for the therapeutic application of inhibiting and activating TAM modulators, particularly with respect to their target specificity and the avoidance of possible treatment side effects in the immune, visual, and reproductive systems.

## Materials and methods

### Antibodies and reagents

Antibodies used were as follows: anti-Tyro3 (Santa Cruz, Dallas, TX, sc-1095, R&D Systems, Minneapolis, MN, AF759), anti-Axl (Santa Cruz sc-1097, R&D Systems AF854), anti-Mer (R&D Systems AF591), anti-HA (Covance, Princeton, NJ, MMS-101P, Roche, Mannheim, Germany 11 867 423 001), anti-Gas6 (R&D Systems AF986), anti-Akt (Cell Signaling, Beverly, MA, 4691), anti-Gapdh (Millipore, Billerica, MA, MAB374), anti-cleaved caspase 3 (Cell Signaling 9661), and anti-phosphotyrosine (Millipore 05–321). Human Pros1 was from Enzyme Research Laboratories, South Bend, IN, and Tet-Express from Clontech, Mountain View, CA.

### Mice

Mice were bred and housed in the Salk Institute Animal Facility in a sterile environment under a 12-hr light/dark cycle. All experiments and procedures were conducted according to the guidelines established by the Institutional Animal Care and Use Committee (IACUC). The *Tyro3*−/−, *Axl*−/−, and *Mertk*−/− mutants (*Lu et al., 1999*), the *Gas6*−/− mutants (*Angelillo-Scherrer et al., 2001*), and the floxed *Pros1* mutants (*Burstyn-Cohen et al., 2012*) were all as described previously. All mice used in this paper were on a pure C57Bl/6 background, with two exceptions: the *Tyro3*−/−*Axl*−/−*Mertk*−/− (TAM TKO) triple mutant mice of *Figure 8—figure supplement 1A* and the young/old comparison of *Figure 8—figure supplement 1B,D*, both of which were on a mixed C57Bl/6 × 129sv background.

### Generation of cell lines

Spontaneously immortalized MEF cell lines from *Tyro3*−/−*Axl*−/−*Mertk*−/− and *Tyro3*−/−*Mertk*−/− mutant mice were generated following a standard 3T3 protocol (*Xu, 2005*). Briefly, E13.5–15.5 embryos were isolated from a pregnant female of each genotype, and embryo body was subject to manual dissociation and incubation with trypsin to isolate single cells. Proliferation of isolated cells was monitored for

15–25 passages, and a growth curve was calculated for each cell passage. Immortalized MEFs were subsequently frozen and utilized for experiments.

C-terminal HA-tagged Tyro3 or Axl cDNA was sub-cloned into pMX IRES retroviral vector. Retrovirus was produced in Phoenix Ampho cell lines and used to transduce TAM TKO cells. For Tet-inducible Mer cell lines, C-terminal HA-tagged Mertk cDNA was sub-cloned into a pRetroX-Tre3G vector and co-transfected with pAmpho env expression vector into a GP2-293 cell for retrovirus production according the manufacturer's protocol (Clontech). Clonal populations were generated following antibiotic selection and receptor levels normalized by quantitative immunoblotting using the HA tag.

## Generation and purification of recombinant TAM ligands

cDNAs encoding either full-length mouse Gas6/Pros1 or Gas6/Pros1 lacking the Gla domain were sub-cloned into pCep4 vector (Invitrogen, Carlsbad, CA). PCR primers introduced a C-terminal $His_6$ tag for purification and the constructs were transfected into HEK293 cells. Cells were grown to confluency in DMEM supplemented with 10% FBS, 0.25 mg/ml G418, 100 µg/ml hygromycin, and Pen/Strep and subsequently switched to serum-free Pro CDMa medium (Lonza, Walkersville, MD) supplemented with 10 µM vitamin K2 (Sigma, St. Louis, MO) and Pen/Strep for 72 hr. Conditioned medium was passed through a 0.22 µm filter, affinity purified through a nickel-NTA resin, and further purified on an anion exchange column (Mono Q 5/5; GE Healthcare, Pittsburgh, PA) in 20 mM Tris–HCl, pH 8.0 using a sodium chloride gradient.

## $^{125}$I-rmGas6 binding assays

Recombinant full-length mouse Gas6 was $^{125}$I-labeled by the iodogen method according to the manufacturer's protocol (Pierce, Rockford, IL). Immortalized $Tyro3^{-/-}Mer^{-/-}$ MEFs were plated in a 12-well dish in triplicate and subsequently pre-starved in 3% FBS in DMEM overnight until cells reach confluency. Cells were placed in starvation medium for 3 hr before binding assays were performed. For binding experiments, cells were incubated with $^{125}$I-labeled full-length rmGas6 with increasing concentrations of either unlabeled full-length rmGas6 or unlabeled Gla-less Gas6 for 3 hr on ice. Cells were washed and subsequently lysed overnight in 0.5 M NaOH. Radioactive content of the cell lysates was counted in Opti-Fluor (Perkin Elmer, Waltham, MA) using a scintillation counter (Beckman Coulter, Brea, CA, LS6500). The half-life of the displacement curves was determined by fitting the curve with Prism software (Graphpad).

## Immunocytochemistry

Cells were plated on glass coverslips, pre-starved overnight in 3% FBS/DMEM, and subsequently incubated in serum-free medium for 3 hr prior to treatment. Coverslips were briefly washed with 1× PBS, fixed for 15 min with 4% paraformaldehyde (PFA)/PBS, incubated in blocking buffer (3% BSA/0.05% saponin/PBS), primary antibody, and fluorophore-conjugated secondary antibody diluted in blocking buffer. For live labeling for surface receptor expression (*Figure 1B* and *Figure 6—figure supplement 1*), coverslips were transferred to chilled serum-free cell culture medium supplemented with anti-Tyro3 or anti-Axl antibodies (R&D Systems) on ice for 30 min. Coverslips were briefly washed with ice cold PBS, fixed for 15 min in 4% PFA/PBS, incubated with a fluorophore-conjugated secondary antibody, and counterstained with Hoechst solution. For live labeling for Gas6 binding assays (*Figure 4D*), coverslips were transferred to chilled serum-free cell culture medium with and without TAM ligand or EDTA, respectively, at 4°C for 90 min. Coverslips were briefly washed with ice cold PBS, fixed for 15 min in 4% PFA/PBS, and incubated with primary antibody, fluorophore-conjugated secondary antibody, and counterstained with Hoechst solution. Following washing with 0.1% Tween-20/PBS, coverslips were mounted using Fluoromount G (Electron Microscopy Sciences) and visualized with a Zeiss LSM 710 Laser Scanning Confocal Microscope.

## Tissue culture, immunoprecipitation, and western blotting

MEFs were grown in Dulbecco's Modified Eagle's Medium supplemented with 10% FBS, and Pen/Strep, and selection antibiotic. Prior to stimulation with ligand, cells were pre-starved overnight in DMEM supplemented with 3% FBS and Pen/Strep and subsequently incubated in starvation conditions for 3 hr.

For BMDM cultures, bone marrow was isolated from tibiae and femurs of 6- to 12-week old C57Bl/6 mice according to the guidelines of the International Animal Care and Use Committee (IACUC). Bones were flushed with DMEM supplemented with 10% FBS and Pen/Strep, spun down at 350 × *g* for 6 min, incubated with ACK lysis buffer for 1 min, and hematopoietic progenitors

incubated with DMEM supplemented with 10% FBS, 30% L929 conditioned medium, Pen/Strep, and glutamax. Cells were supplemented with fresh medium on day 3 and cells re-plated on day 7 in DMEM supplemented with 10% FBS and Pen/Strep for experiments. For activation assays, cells were placed under starvation conditions supplemented with 100 nM dexamethasone and Pen/Strep overnight.

Following stimulation with TAM ligands at the indicated concentrations and time periods, cells were washed in PBS and lysed in lysis buffer [50 mM HEPES, pH 7.5, 150 mM NaCl, 1 mM EDTA, 1 mM EGTA, 10% glycerol, 1% Triton X-100, 25 mM NaF, 10 μM $ZnCl_2$, and protease and phosphatase inhibitors (Roche, Sigma)] for 30 min on ice. For immunoprecipitation, cell lysates were incubated with anti-HA high affinity (Roche), anti-Mer (R&D System), or anti-Tyro3 (Santa Cruz) antibodies, and either protein A/protein G-Sepharose conjugated beads (Life Technologies). Immunoprecipitates were washed 4× with HNTG buffer (20 mM HEPES, pH 7.5, 150 mM NaCl, 0.1% Triton X, 10% glycerol, 1 mM $Na_3VO_4$), separated by SDS-PAGE, transferred to PVDF membrane (Millipore), incubated with primary antibodies in blocking buffer (1% casein block in PBS) (Biorad), and western blots were developed using an Odyssey Gel Imaging System (Licor).

### Histology and immunohistochemistry

Mice were collected at 12–14 weeks and were anesthetized with 2.5% avertin in saline. Mice were perfused with 20 U/ml heparin/PBS and subsequently with 4% PFA in PBS. Eyes were processed as previously described (*Burstyn-Cohen et al., 2012*). Testes were collected, immersion fixed overnight at 4°C, infiltrated with 30% sucrose/PBS overnight at 4°C, and flash frozen in TBS tissue freezing medium. 10 μm serial sections were cut for light microscopy studies and immunohistochemistry, respectively. Sections were air dried overnight at room temperature and subsequently frozen at −80°C.

For cCasp3 staining, sections were washed in PBS, incubated in 0.3% $H_2O_2$ in PBS to block endogenous peroxidase activity, blocked in 3% BSA in PBS, and incubated in anti-cCasp3 (Cell Signaling) diluted in blocking buffer. DAB staining was performed with an ABC Vectastain Kit (Vector Labs) and Peroxidase Substrate Staining Kit (Vector Labs). Following washing steps in PBS + 0.1% Tween-20, slides were counterstained with hematoxylin for 1 min and mounted using VectaMount (Vector Labs). For quantitation, cCasp3-positive cells and tubule numbers from four testis cross-sections per animal were counted and averaged per animal. Error bars represent standard error of the mean from three independent animals.

### Flow cytometry

Cells were plated in 6-cm dishes and when cells reached confluency, they were pre-starved in 3% FBS/DMEM overnight and then incubated in starvation medium for 3 hr before experiment. Supernatant was collected for analysis, and adherent cells were washed with PBS, incubated with 0.25% trypsin briefly, and collected in FACS buffer (0.5% BSA/0.1% Na azide/PBS supplemented with either 2 mM CaCl2 or 10 mM EDTA). Supernatant and cell samples were incubated with pSiva-IANBD and Propidium iodide (PI) (Imgenex) for 30 min on ice. Samples were sorted on a Becton-Dickinson LSR II (Salk Institute Flow Cytometry Core Facility) and data analyzed using FloJo software (Treestar).

### Gas6 injections in vivo

Male and female 13- to 15-week old mice were injected intravenously (retro-orbital injection, 100 μl final volume) with 30 μg full-length recombinant mouse Gas6 (in saline), Gla-less recombinant Gas6, or saline as a control. Mice were sacrificed at 30 min post-injection and the spleens split in half: one half was snap frozen for biochemical analysis and the second half was fresh frozen in OCT freezing medium, as indicated in *Figure 5*.

### Acknowledgements

We thank Joseph Hash for excellent technical support and members of the Lemke and Schlessinger labs for helpful discussions. This work was supported by grants from the National Institutes of Health (R01 AI077058, R01 AI101400, and R01 NS085296 to GL, and P30CA014195 to the Salk Institute) and the Leona M and Harry B Helmsley Charitable Trust (#2012-PG-MED002 to the Salk Institute), by the Nomis, H N and Frances C Berger, Fritz B Burns, and HKT Foundations, by Frederik Paulson and Françoise Gilot-Salk, and by post-doctoral fellowships from the Leukemia and Lymphoma Society and the Nomis Foundation (to EDL), the Human Frontiers Science Program (to AZ), and the Marie Curie International Outgoing Fellowship Program (to PGT).

# Additional information

## Competing interests

IL: is a shareholder in Kolltan Pharmaceuticals. JS: is a shareholder in Kolltan Pharmaceuticals. GL: is a shareholder in Kolltan Pharmaceuticals. The other authors declare that no competing interests exist.

## Funding

| Funder | Grant reference number | Author |
|---|---|---|
| National Institutes of Health | R01 AI077058, R01 AI101400, R01 NS085296, P30 CA014195 | Greg Lemke |
| Leona M. and Harry B. Helmsley Charitable Trust | 2012-PG-MED002 | Greg Lemke |
| Leukemia and Lymphoma Society | | Erin D Lew |
| Nomis Foundation | | Erin D Lew, Greg Lemke |
| Fritz B. Burns Foundation | | Greg Lemke |
| Haeyoung Kong Tang Foundation | | Greg Lemke |
| H.N and Frances C. Berger Foundation | | Greg Lemke |
| Human Frontier Science Program | | Anna Zagórska |
| European Commission | Marie Curie International Outgoing Fellowship Program | Paqui G Través |

The funders had no role in study design, data collection and interpretation, or the decision to submit the work for publication.

## Author contributions

EDL, Conception and design, Acquisition of data, Analysis and interpretation of data, Drafting or revising the article; JO, PGB, IL, Conception and design, Acquisition of data, Analysis and interpretation of data; AZ, PGT, Acquisition of data, Analysis and interpretation of data, Drafting or revising the article; JS, Conception and design, Drafting or revising the article, Contributed unpublished essential data or reagents; GL, Conception and design, Analysis and interpretation of data, Drafting or revising the article

## Author ORCIDs

Paqui G Través, http://orcid.org/0000-0001-5749-8426

## Ethics

Animal experimentation: This study was performed in strict accordance with the recommendations in the Guide for the Care and Use of Laboratory Animals of the National Institutes of Health. All of the animals were handled according to approved Institutional Animal Care and Use Committee (IACUC) protocol of the Salk Institute, Animal Use Protocol No. 11-00051, approval date of record 3 June 2014.

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
