## [Decision Letter]

Thank you for sending your work entitled “Differential TAM receptor-ligand-phospholipid interactions delimit differential TAM bioactivities” for consideration at *eLife*. Your article has been favorably evaluated by Charles Sawyers (Senior editor), a Reviewing editor, and 3 reviewers.

The Reviewing editor has assembled the following comments to help you prepare a revised submission.

A common complaint is the use of human S protein in much of the work instead of the murine S protein. This seems to be a big problem in the reviewers' minds and will need to be addressed. Either much of the work will have to be repeated with the mouse protein, or some disclaimers will have to be injected into the paper that convince us that this is not an important issue.

There are several complaints about quantification of the responses, titration of stimuli, error bars, and saturation of responses. The points seem clearly laid out. I think the complaints have merit and need to be addressed, either experimentally or verbally. Good quantification, with error bars, is needed to validate the claims drawn from the data. This would seem to be a major part of the paper and so must be dealt with.

*Reviewer 1 major comments*:

The paper describes a large body of work to address important questions related to the TAM receptors and their ligands. The authors have created unique reagents and the results obtained are very interesting and valid. The paper addresses questions that have previously been studied by others, but the unique set of reagents make it possible for them to give clear answers. The work is of high quality and the conclusions are valid. I have the following concerns the authors need to address:

The authors have used human protein S in the majority of the experiments as it showed to be more potent than murine protein S as a TAM-activator. In a study wanting to compare the physiological role of TAMs and their ligands in a defined setting, this is not really relevant. The study would benefit from using murine protein S throughout the study.

In the experiments testing the thrombin-cleavage-dependent effects of protein S, I am not convinced the protein S has been fully processed. Have the authors verified that the thrombin cleavage is complete (e.g. using a GLA-antibody on a western blot)? As the authors state that only half the physiological amounts of protein S is needed in vivo to activate TAMs, a partial thrombin cleavage might not show strong effects in a stimulation experiment. Likewise, it would be good to use in addition a lower protein S concentration for the stimulation to see a dose dependence, as well as to study also Mer phosphorylation in the same setup. In addition, it is not clear to me why the authors have used mouse proteinS for this experiment whereas most other are performed with human protein S.

The authors discuss the role of ptdser in TAM activation by Gas6 and suggest that ptdser is always needed for optimal TAM activation. An experiment on receptor phosphorylation in the presence of an excess of liposomes with and without ptdser could easily show if there is an added effect of circulating ptdser. In addition, it is not clear whether the authors suggest that ptdser-exposure on the same cell as TAM-expression or ptdser in circulation/on interacting cells is the crucial factor.

The authors claim that sAxl is released from the cell surface upon Axl activation. This has not been clearly shown and should not be addressed as a fact. An experiment studying sAxl release into the cell culture supernatant upon Gas6-mediated Axl stimulation could give more support to this claim.

*Reviewer 2 major comments*:

Lew et al. describe an interesting set of studies of ligand engagement and activation in the TAM family, clearing up several important issues and providing some valuable observations. Using purified ligands and MEFs engineered to express single TAMs (a clear and elegant system for these studies), they show that Gas6 functions as a ligand for Tyro3, Axl, and Mer. They argue that Axl is most potently activated (see point1 and 2 below). On the other hand, the authors show that Pros1 activates Tyro3 and Mer, but argue that it does not activate Axl ( see point 1 and 3 below). They further show that the Gla domain is important for activation – but not necessarily for receptor binding, and Axl appears most dependent on the presence of the Gla domain in Gas6 (see point 4 below). The observed differences appear to track with the Ig domains of the receptors. Finally, the data supporting Pros1/Mer signaling in retinal phagocytosis are quite compelling, and isolate that signaling axis impressively in an *in vivo* setting, and seem to support separate signaling functions of each receptor/ligand pair.

The systems used to tease out the signaling relationships here are excellent, and the data very informative for a poorly understood and interesting system. As a result, the manuscript should be of wide interest. There are a few important problems, though, most notably the tendency to draw quantitative conclusions about relative potencies from dose-response curves that are not saturated, and from data presented without proper quantitation or assessment of error (see point 1). The conclusions seem like they are broadly correct, but these biochemical data need to be presented in a way that can be objectively evaluated. Points 1-3 focus on this issue, which really needs to be addressed properly before publication should be considered.

Major points:

1) There is a general concern throughout the manuscript that interpretations about relative potencies (and half-max ligand concentrations) are made without quantitation or (therefore) error bars – and it is not stated how many repeats of each analysis were done. This issue resurfaces in several of the comments below that concern quantitative interpretations. Using Licor to follow tyrosine phosphorylation, results of quantification of 3 or more repeats needs to be shown, with error, for comparison of potencies etc.

2) The dose-response assessment in Figure 2 does not really show that rmGas6 is a 'more effective' ligand for Axl than Tyro3. Neither assessment reaches saturation of receptor phosphorylation (see point 1 above), and the trends with ligand concentration look very similar as far as they go. If the two receptors saturate at different pY/4G10 signals, the half maximal values may very well be the same (with no higher activation for Axl). There are similar problems with many of the dose-response curves in the paper (although signal may be saturated in rmGas6 activation of Tyro3 in Figure 2, and in Figures 3 and 5), making comparisons difficult.

3) Given the background signal in Figure 2 (compared with 2E), it is difficult to be completely convinced that hPros1 activates Mer but not Axl. There is clearly some Mer activation - although it remains unclear what hPros1 concentration is required to saturate. For Axl in Figure 2, comparison of the 5, 10, and 50nM lanes suggest some activation - but the 100nM lane argues otherwise (how many times was this done - the authors do not say). It is difficult to know whether this is a problem with just that sample. Comparing Pros1 activation of Mer in Figure 2 (claimed lack of) Axl activation in Figure 2 also reduces confidence. The data need to be quantitated in order to make the quantitative comparison made here (as done for Mer in Figure 2 but not for Axl in Figure 2). Error bars also need to be provided for the Licor-based quantitation – with statement of how many repeats were performed – if the case for Pros1 activation of Mer but not Axl is to be made convincingly.

4) One has to wonder how physiologically significant is the difference between Axl and the other TAMs in terms of Gla domain dependence. Whereas Gla-less Gas6 seems to lose efficacy for Axl, removal of the Gla domain still reduces potency perhaps 20-50 fold for Mer and Tyro3. It would be helpful for the authors to put this into physiological context. What is the functional difference between 'dead' and 20-50 fold impaired for Gas6? Is there any?

5) Was cell surface expression of ECDTIgITKA confirmed, to exclude the trivial possibility that it is not accessible to hPros1 in Figure 5?

*Reviewer 3 major comments*:

Lew et.al. investigated the mechanisms of signaling mediated by Gas6 and protein S via interaction(s) with TAM receptors (TYRO3, Axl, Mer). Cell lines derived from TAM knockout mice enabled them to analyze the ability of Gas6 and protein S to elicit signaling from each type of TAM receptor by ectopically expressing one type of the receptors. They found that Gas6 can elicit signaling through all TAM receptors, while protein S can only do so through Mer and TYRO3. Using chimeric receptors, they demonstrated that the extracellular Ig domains of the receptors determine the ability of the receptors to elicit signaling in response to protein S. They also found that interactions between the Gla domain of protein S/Gas6 and phosphatidylserine are necessary for optimal signaling. They analyzed the roles of protein S, Gas6, and TAM receptors in phagocytosis in the retina and testis, using various types of knockout mice. Based upon analysis of knockout mouse phenotypes and their data from *in vitro* analysis of the signaling mechanisms, they claim that Mer and protein S play a predominant role in phagocytosis in the retina and testis. This reviewer sees two major concerns regarding experimental design used in this study:

1) The authors used human protein S in studies investigating the mechanisms of signaling in murine TAM receptors (Figures 2, 5 and 6). Because the activity and specificity of protein S and Gas6 could be species-specific (ref. Hafizi and Darlback, FEBS J., 2006), it is not clear whether all the data acquired with human protein S are applicable to the signaling mechanisms of murine protein S. In addition, preparation of human protein S differed from that of murine protein S and Gas6; human protein S was purified from human serum, and both murine and human protein S were generated as His-tag-conjugated recombinant proteins. Therefore, comparison of human protein S and murine Gas6 abilities to induce signaling can be influenced by many factors, including purification methods, presence of tag sequence, and efficiency and types of post-translational modifications.

Because the data that compare signaling activity and specificity of murine protein S and Gas6 are very important to properly interpret the mouse *in vivo* data, most of the *in vitro* experiments investigating signaling mechanisms of protein S should be performed using murine protein S.

2) The authors claim that PtdSer exposed on a small fraction of cultured cells binds to the Gla domain of protein S and Gas6, and this binding is necessary for optimal signaling via TAM receptors. To clearly show that PtdSer exposed on minor cell populations plays a critical role in Gas6/protein S signaling, it will be necessary to demonstrate that induction of PtdSer exposure and elimination of exposed PtdSer on cells affect the magnitude of signaling by protein S/Gas6.

---

## [Author Response]

*A common complaint is the use of human S protein in much of the work instead of the murine S protein. This seems to be a big problem in the reviewers' minds and will need to be addressed. Either much of the work will have to be repeated with the mouse protein, or some disclaimers will have to be injected into the paper that convince us that this is not an important issue*.

As detailed below, we have done both. We have included significant additional data using recombinant mouse Protein S as a ligand for the TAMs, and have also cited recently published (e-published ahead of print) work in which human and mouse Protein S were found to be equivalent with respect to their ability to activate a hybrid receptor composed of either the Tyro3 or Mer extracellular domain linked to the cytoplasmic domain of the R1 chain of the IFNγ receptor. (Activation of the hybrid chimeras was monitored by STAT1 phosphorylation.) In general, we have used the purified human protein only because it is (for unknown reasons) more stable biochemically over time than the recombinant mouse protein that we produce and purify. However, we now show that both purified human and recombinant mouse Protein S have the same receptor specificity.

*There are several complaints about quantification of the responses, titration of stimuli, error bars, and saturation of responses. The points seem clearly laid out. I think the complaints have merit and need to be addressed, either experimentally or verbally. Good quantification, with error bars, is needed to validate the claims drawn from the data. This would seem to be a major part of the paper and so must be dealt with*.

We have addressed issues with respect to quantitation and reproducibility, which primarily relate to Figure 2, as outlined below in our responses to the individual reviews.

Reviewer 1 major comments:

*[…] The work is of high quality and the conclusions are valid. I have the following concerns the authors need to address*:

*The authors have used human protein S in the majority of the experiments as it showed to be more potent than murine protein S as a TAM-activator. In a study wanting to compare the physiological role of TAMs and their ligands in a defined setting, this is not really relevant. The study would benefit from using murine protein S throughout the study*.

The human Protein S we have used is a commercial product (Enzyme Research Laboratories) purified from human plasma, whereas the mouse Protein S is a recombinant protein that we have expressed and purified in-house. For unknown reasons, the purified human protein is in our hands slightly more potent and significantly more stable biochemically than the recombinant mouse protein.

This is the main reason that we have used it for many of our experiments. However, both the human and mouse proteins show the same specificity and are active ligands for mouse Tyro3 and Mer. We have now indicated this in the Results section of the revised paper. As suggested by Reviewer 1, we have also included additional results obtained with our recombinant mouse Protein S, in Figure 2, panels B, E and Figure 2—figure supplement 1. We have also cited a recent paper from Tsou *et al.* (*J. Biol. Chem*., in press, pii: jbc.M114.569020) in which mouse Protein S and human Protein S were compared with respect to their ability to induce STAT1 phosphorylation upon activation of a hybrid receptor composed of either the Tyro3 or Mer extracellular domain linked to the cytoplasmic domain of the R1 chain of the interferon-γ receptor. Human and mouse Protein S were found to be equivalently good ligands for both mouse Mer/γR1 and mouse Tyro3/γR1, as monitored by the activation of STAT1.

*In the experiments testing the thrombin-cleavage-dependent effects of protein S, I am not convinced the protein S has been fully processed. Have the authors verified that the thrombin cleavage is complete (e.g. using a GLA-antibody on a western blot)? As the authors state that only half the physiological amounts of protein S is needed in vivo to activate TAMs, a partial thrombin cleavage might not show strong effects in a stimulation experiment. Likewise, it would be good to use in addition a lower protein S concentration for the stimulation to see a dose dependence, as well as to study also Mer phosphorylation in the same setup. In addition, it is not clear to me why the authors have used mouse proteinS for this experiment whereas most other are performed with human protein S*.

Reviewer 1 is correct on each of these points. The problem that we face is that if we allow the thrombin incubation to run long enough for all of the Protein S to be cleaved, all of its bioactivity is lost. This is in spite of the fact that in intact serum, in which almost of the Protein S is thrombin-cleaved, the resident Protein S is a strong Tyro3 ligand. (It is important to remember that the objective of the experiment was test if cleavage is required to *activate* Protein S, which is apparently not the case since rmPros1 is active *before* thrombin cleavage.) Since these results are equivocal and not relevant to the main points of the paper, we have removed this supplemental panel from the revised paper.

*The authors discuss the role of ptdser in TAM activation by Gas6 and suggest that ptdser is always needed for optimal TAM activation. An experiment on receptor phosphorylation in the presence of an excess of liposomes with and without ptdser could easily show if there is an added effect of circulating ptdser*.

We published a similar experiment last year (Bhattacharyya *et al.*, *Cell Host & Microbe* 14(2): 136-47, Figure 3) in which we titered increasing concentrations of Gas6 and Protein S, in the absence or presence HIV-1-derived retroviruses that display PtdSer on the surface of their membrane envelopes. (The high level of surface PtdSer on HIV-1 allows the virus to be purified using Annexin V affinity methods; e.g., Callahan *et al.*, *J. Immunol*. 170: 4840.) We showed that the presence of the PtdSer-containing membranes dramatically shifted the dose-response curves (to lower protein concentration) for both Mer activation by Protein S and Axl activation by Gas6. In addition, the paper referenced as ‘Zagórska *et al.* submitted’ is now in press at *Nature Immunology*.

This paper contains a similar experiment using PtdSer-expressing apoptotic cells, and reaches the same conclusion. We have cited these experiments, and highlighted their significance in the Results section of the revised paper.

*In addition, it is not clear whether the authors suggest that ptdser-exposure on the same cell as TAM-expression or ptdser in circulation/on interacting cells is the crucial factor*.

In the case of the two settings that we analyze genetically in the paper – the phagocytosis of the distal ends of photoreceptor outer segments by RPE cells in the eye, and the phagocytosis of apoptotic germ cells by Sertoli cells in the testis – the PtdSer is unambiguously expressed on the membrane surface of an apposed cell. Without question, this is also the case for the phagocytosis of apoptotic cells by macrophages. In all of these settings, the phagocyte has the TAM receptor, which is TAM-ligand-bridged to the engulfment target, which displays the PtdSer. We do not exclude the possibility the PtdSer-rich membranes might bind ligand and thereby activate a TAM receptor in the same cell, but we envision that the geometry of this signaling interaction will make these situations unusual.

*The authors claim that sAxl is released from the cell surface upon Axl activation. This has not been clearly shown and should not be addressed as a fact. An experiment studying sAxl release into the cell culture supernatant upon Gas6-mediated Axl stimulation could give more support to this claim*.

Related experiments have been published. The Wilhelm *et al.* paper that we cite (*J. Neurochem.*, 2008, 107: 116) demonstrates that Axl cleavage by matrix metalloproteinases occurs subsequent to Axl activation (tyrosine phosphorylation), and can be inhibited by the broad-spectrum MMP inhibitor GM6001. We also cite (in the Discussion section of the revised paper) results from our in-press *Nature Immunology* paper, in which we show that antibody-induced Axl activation in tissues *in vivo* is accompanied by Axl proteolytic cleavage, and that the extent of cleavage is correlated with the level of Axl activation (more tyrosine phosphorylation, more cleavage).

Reviewer 2 major comments:

*[The] manuscript should be of wide interest. There are a few important problems, though, most notably the tendency to draw quantitative conclusions about relative potencies from dose-response curves that are not saturated, and from data presented without proper quantitation or assessment of error (see point 1). The conclusions seem like they are broadly correct, but these biochemical data need to be presented in a way that can be objectively evaluated. Points 1-3 focus on this issue, which really needs to be addressed properly before publication should be considered*.

*Major points*:

*1) There is a general concern throughout the manuscript that interpretations about relative potencies (and half-max ligand concentrations) are made without quantitation or (therefore) error bars – and it is not stated how many repeats of each analysis were done. This issue resurfaces in several of the comments below that concern quantitative interpretations. Using Licor to follow tyrosine phosphorylation, results of quantification of 3 or more repeats needs to be shown, with error, for comparison of potencies etc*.

We have addressed these concerns with respect to activation profiles as detailed in our responses to the specific points of Reviewer 2 that follow. Although most of the immunoblots in the paper are indeed Licor blots, the panels showing Mer activation have a signal that is too low to be detected using Licor and these had to be developed using ECL (Figure 2, panels G, H, and supplement 1, as indicated in the figure legends). This is due to the lower Mer expression that we discuss in the Results section. Therefore, we could not use universal quantification for all the receptors. Instead we have shown multiple representative examples of our results. For example, in the revised Figure 2, the reader can now see three separate dose-response results for rmGas6 activation of Tyro3, all of which reach saturation. In the revised paper, we have also moderated the discussion of half-max ligand concentrations.

*2) The dose-response assessment in*
Figure 2
*does not really show that rmGas6 is a 'more effective' ligand for Axl than Tyro3. Neither assessment reaches saturation of receptor phosphorylation (see point 1 above), and the trends with ligand concentration look very similar as far as they go. If the two receptors saturate at different pY/4G10 signals, the half maximal values may very well be the same (with no higher activation for Axl). There are similar problems with many of the dose-response curves in the paper (although signal may be saturated in rmGas6 activation of Tyro3 in*
Figure 2*, and in*
Figures 3 and 5*), making comparisons difficult*.

Reviewer 2 has a valid point here. We chose MEF lines that express approximately equal levels of Tyro3 and Axl (based on expression of the HA tag) and we can often detect Axl phosphorylation at slightly lower Gas6 levels than we can detect Tyro3 activation. However, Gas6 is indeed a strong ligand for both Axl and Tyro3. (Compare, for example, new panels 2B first 8 lanes, 2D left 7 versus right 7 lanes, and 2E left 7 lanes. All of these new panels reach saturation.) We have revised the text accordingly, in the Results section. Perhaps the most salient point of the Figure 2 analysis is that Axl can *only* be activated by Gas6.

*3) Given the background signal in*
Figure 2
*(compared with 2E), it is difficult to be completely convinced that hPros1 activates Mer but not Axl. There is clearly some Mer activation - although it remains unclear what hPros1 concentration is required to saturate. For Axl in*
Figure 2*, comparison of the 5, 10, and 50nM lanes suggest some activation - but the 100nM lane argues otherwise (how many times was this done - the authors do not say). It is difficult to know whether this is a problem with just that sample. Comparing Pros1 activation of Mer in*
Figure 2
*(claimed lack of) Axl activation in*
Figure 2
*also reduces confidence. The data need to be quantitated in order to make the quantitative comparison made here (as done for Mer in*
Figure 2
*but not for Axl in*
Figure 2*). Error bars also need to be provided for the Licor-based quantitation* – *with statement of how many repeats were performed* – *if the case for Pros1 activation of Mer but not Axl is to be made convincingly*.

As indicated above in our response to Reviewer 1, we have added new additional data – in Figure 2, panels B and E and Figure 2—figure supplement 1 – for titration of recombinant mouse Pros1 against both Axl and Tyro3 expressed in isolation in MEFs, and Mer expressed in Dex-treated macrophages. We have also added new panels F and H to Figure 2, which show that hPros1 is able to activate Tyro3 and Mer but not Axl. The ‘some activation’ of Axl by Pros1 mentioned by Reviewer 2 reflects lower molecular weight bands that are seen specifically in α-pY blots for Axl-expressing MEFs only when Axl is *not* activated by exogenous Gas6 – they consistently disappear when Axl is activated, and they do not increase in intensity with increasing concentrations of either rmPros1 or hPros1. (See Figure 2 panel D, lane 8; panel E, lanes 1 and 8-11; and panel F, lanes 8-14.) We have clarified the nature of these bands (which may reflect a low basal level of phosphorylation due to MEF production of endogenous Gas6) in the legend to the revised Figure 2. These new data are consistent with our original data, and clearly demonstrate that neither hPros1 nor rmPros1 can activate Axl. They are also consistent with the in-press work of Tsou *et al.* (*J. Biol. Chem*., in press, pii: jbc.M114.569020) cited above, in which mouse Pros1 was found to be able to activate STAT1 phosphorylation downstream of hybrid receptors composed of the ectodomain of either Tyro3 or Mer linked to the cytoplasmic domain of the R1 chain of the IFN-γ receptor, but *not* to activate a similar hybrid receptor containing the Axl ectodomain. This is now discussed in the Results section.

Finally, in our in-press *Nature Immunology* paper, we show that phagocytosis of apoptotic cells by macrophages that exclusively express Mer can be stimulated by both Gas6 and Protein S, whereas phagocytosis in macrophages that express exclusively Axl can only be stimulated by Gas6. In concert, all of this new work overwhelmingly argues that Axl is activated by Gas6 but not by Protein S.

*4) One has to wonder how physiologically significant is the difference between Axl and the other TAMs in terms of Gla domain dependence*. *Whereas Gla-less Gas6 seems to lose efficacy for Axl, removal of the Gla domain still reduces potency perhaps 20-50 fold for Mer and Tyro3. It would be helpful for the authors to put this into physiological context. What is the functional difference between 'dead' and 20-50 fold impaired for Gas6? Is there any?*

This is an interesting possibility, but one that we can’t address definitively.

It is indeed possible that a 20-50-fold reduction in potency might mean that the Gla-less Gas6 is effectively inert as a Tyro3 or Mer ligand *in vivo*, but this will depend on local ligand concentration, clustering, and other variables. However, the key point we want to make is that in this assay system, there is clearly a difference between Tyro3/Mer and Axl – this time in terms of the ability of Glaless Gas6 to activate them.

With respect to the inability of Gla-less Gas6 to activate Axl, we have also added a new set of *in vivo* experiments and a new Figure 5 to the revised paper. We believe that these new experiments extend the findings of Figure 3 and Figure 4 in a very powerful way. We injected (intravenously) either full-length or Gla-less recombinant mouse Gas6 into *Gas6-/-* mice, and then monitored the co-localization of the injected proteins with red pulp macrophages in the spleen 30 minutes after injection. (These macrophages are Axl-expressing.) We found that IV-injected full-length and Gla-less Gas6 bound equally well to these cells.

However, when we examined Axl activation (tyrosine phosphorylation) in these same macrophages, we found that the full-length ligand induced strong activation, while the Gla-less Gas6 produced no Axl activation at all – equivalent to an injection of saline. We assume that the PtdSer required for Axl activation by full-length injected Gas6 comes from the blood, which is full of PtdSer-displaying microparticles (derived from platelets, erythrocytes, leukocytes, and endothelial cells). These dramatic new results are presented in the Results section of the revised paper.

*5) Was cell surface expression of ECDTIgITKA confirmed*, *to exclude the trivial possibility that it is not accessible to hPros1 in*
Figure 5*?*

Yes – surface expression was confirmed. We have added a figure supplement to Figure 6 to document this.

Reviewer 3 major comments:

[…] This reviewer sees two major concerns regarding experimental design used in this study:

*1) The authors used human protein S in studies investigating the mechanisms of signaling in murine TAM receptors (*Figures 2, 5 and 6*). Because the activity and specificity of protein S and Gas6 could be species-specific (ref. Hafizi and Darlback, FEBS J., 2006), it is not clear whether all the data acquired with human protein S are applicable to the signaling mechanisms of murine protein S. In addition, preparation of human protein S differed from that of murine protein S and Gas6; human protein S was purified from human serum, and both murine and human protein S were generated as His-tag-conjugated recombinant proteins. Therefore, comparison of human protein S and murine Gas6 abilities to induce signaling can be influenced by many factors, including purification methods, presence of tag sequence, and efficiency and types of post-translational modifications*.

*Because the data that compare signaling activity and specificity of murine protein S and Gas6 are very important to properly interpret the mouse in vivo data, most of the in vitro experiments investigating signaling mechanisms of protein S should be performed using murine protein S*.

As indicated above (please see our response to Reviewer 2, point 3), we have now included additional data with recombinant mouse Protein S in revised Figure 2 and its supplement (five new panels in all). It is also important to emphasize that the genetic studies we present in the retina (revised Figure 7), together with those we have previously published in Burstyn-Cohen et al., *Neuron* 76: 1123-32 (2012), unambiguously demonstrate that mouse Protein S functions as a ligand for mouse Mer.

*2) The authors claim that PtdSer exposed on a small fraction of cultured cells binds to the Gla domain of protein S and Gas6, and this binding is necessary for optimal signaling via TAM receptors. To clearly show that PtdSer exposed on minor cell populations plays a critical role in Gas6/protein S signaling, it will be necessary to demonstrate that induction of PtdSer exposure and elimination of exposed PtdSer on cells affect the magnitude of signaling by protein S/Gas6*.

As discussed above, we have addressed this point in multiple ways. We show that a Gas6 variant that lacks its Gla domain and cannot bind PtdSer binds to Axl with wild-type affinity but is incapable of activating the receptor either in vitro or *in vivo* (Figures 3 and 4 and the new Figure 5). We have shown that the inclusion of PtdSer-containing membranes, from either apoptotic cells or enveloped viruses, shifts the ligand dose-response activation profiles for both Axl and Mer to lower ligand concentrations (Bhattacharyya et al. *Cell Host & Microbe* 14: 136-47 (2013); Zagórska et al., *Nature Immunology*, in press (October 2014), discussed in the Results section). We show that blocking PtdSer with annexin V inhibits Tyro3 activation (Figure 4) and that depletion of calcium ions that are required for PtdSer-Gla domain interaction inhibits Axl activation but not binding by full-length Gas6 (Figure 4). The phagocytosis of apoptotic germ cells by Sertoli cells in the testis and of outer segment membranes by RPE cells in the retina – the two systems we analyze genetically in our paper – have both been previously shown to be entirely dependent on the exposure of PtdSer, and in the case of the retina this exposure cycles during the day and is light-induced (Ruggiero et al., *PNAS* 109: 8145-8 (2012)). This is now discussed in the Discussion section.